**COMMUNICATIONS**

# Genome-wide detection of cytosine methylations in plant from Nanopore data using deep learning

Peng Ni[1,2], Neng Huang[1,2], Fan Nie[1,2], Jun Zhang[1,2], Zhi Zhang[1,2], Bo Wu[3], Lu Bai[4], Wende Liu[4], Chuan-Le Xiao [5✉], Feng Luo [3✉] & Jianxin Wang [1,2✉]

In plants, cytosine DNA methylations (5mCs) can happen in three sequence contexts as CpG, CHG, and CHH (where H = A, C, or T), which play different roles in the regulation of biological processes. Although long Nanopore reads are advantageous in the detection of 5mCs comparing to short-read bisulfite sequencing, existing methods can only detect 5mCs in the CpG context, which limits their application in plants. Here, we develop DeepSignal-plant, a deep learning tool to detect genome-wide 5mCs of all three contexts in plants from Nanopore reads. We sequence *Arabidopsis thaliana* and *Oryza sativa* using both Nanopore and bisulfite sequencing. We develop a denoising process for training models, which enables DeepSignal-plant to achieve high correlations with bisulfite sequencing for 5mC detection in all three contexts. Furthermore, DeepSignal-plant can profile more 5mC sites, which will help to provide a more complete understanding of epigenetic mechanisms of different biological processes.

[1] School of Computer Science and Engineering, Central South University, Changsha 410083, China. [2] Hunan Provincial Key Lab on Bioinformatics, Central South University, Changsha 410083, China. [3] School of Computing, Clemson University, Clemson, SC 29634-0974, USA. [4] State Key Laboratory for Biology of Plant Diseases and Insect Pests, Institute of Plant Protection, Chinese Academy of Agricultural Sciences, Beijing, China. [5] State Key Laboratory of Ophthalmology, Zhongshan Ophthalmic Center, Sun Yat-sen University, #7 Jinsui Road, Tianhe District, Guangzhou, China. ✉email: xiaochuanle@126.com; luofeng@clemson.edu; jxwang@mail.csu.edu.cn

As one of the major DNA methylations, cytosine DNA methylations (5mCs) play important roles in regulating the biological processes of plants[1,2], such as gene expression regulation[3], transposable elements silencing[4], fruit development[5], and stress response[6,7]. In plants, 5mCs can happen in three sequence contexts as CpG, CHG, and CHH (where H = A, C, or T). The methylation levels of 5mCs vary widely among plant species[8]. For example, there are 24% CpG, 6.7% CHG, and 1.7% CHH methylated at read level in *Arabidopsis thaliana*[9,10], while *Beta vulgaris* has 92.5% CpG, 81.2% CHG, and 18.8% CHH being methylated at read level[8]. Three types of 5mCs play different roles in the regulation of biological processes in plants. For example, CpG methylation usually dominates in gene bodies[11]. CHG methylation plays a greater role than CpG methylation in silencing transposons, and CHH methylation is crucial for silencing CpG and CHG-depleted transposons[12]. Therefore, the detection of genome-wide 5mC methylation in all three contexts is important in plants.

Bisulfite sequencing is the most widely used method for profiling 5mC methylation[13]. All three contexts of 5mCs can be detected by bisulfite sequencing. However, due to short-read sequencing, bisulfite sequencing cannot profile 5mCs in the repetitive genome regions. Furthermore, the incomplete conversion and DNA degradation during bisulfite treatment can also lead to the lack of specificity and the loss of sequencing diversity[14,15]. Recently, third-generation sequencing technologies, such as Pacbio single-molecule real-time (SMRT) sequencing and Nanopore sequencing, which can directly sequence the DNA without the conversion or PCR amplification, provide new opportunities for detecting the base modifications in DNA[16,17]. The DNA base modifications can affect the electrical current signals near modified bases in Nanopore sequencing[18] and alter polymerase kinetics during Pacbio SMRT sequencing[17]. Thus, the DNA methylations can be directly detected from native DNA reads of Nanopore and Pacbio SMRT sequencing without extra laboratory techniques, which can avoid DNA degradation and amplification biases. Moreover, the long reads of Nanopore and Pacbio sequencing make it possible to profile methylation in repetitive or low complexity regions[19].

Many methods have been developed to detect 5mCs using either Pacbio or Nanopore reads. Due to the weak effect of 5mC on synthesis kinetics in Pacbio SMRT sequencing, the statistic method using the early version of Pacbio SMRT data to detect 5mCs exists low signal-to-noise ratio problem[17]. Recently, Tse et al. developed a convolutional neural network-based method to detect genome-wide CpG in humans using the new circular consensus sequencing (CCS) reads from Pacbio and the results show a high correlation with bisulfite sequencing[20]. Methods using Nanopore reads for DNA 5mC detection can be classified into three categories. The statistics-based methods, such as Tombo[21], infer DNA methylation by statistically testing current signals between native DNA reads and methylation-free DNA reads. Tombo can detect all types of DNA methylation without a priori knowledge of current signal patterns on specific methylation types. However, Tombo is not reliable at the single nucleotide level and usually has a high false-positive rate[22]. The model-based methods, such as nanopolish[23], signalAlign[24], DeepMod[25], and DeepSignal[26], utilize hidden Markov models or deep neural networks to predict the status of the specific site as modified or unmodified, which achieve high accuracies on 5mC detection in a specific motif, such as CpG or CCWGG (where W = A or T)[27]. The basecalling-based methods, such as Megalodon[28], directly call modified bases using an extended alphabet during basecalling[22]. Megalodon can detect a variety of methylation types, including 5mC in all contexts. However, the capability of Megalodon for CHH and CHG methylation detection is lack of evaluation. To the best of our knowledge, no current method can profile genome-wide 5mCs in all three contexts with acceptable accuracies using third-generation sequencing data.

Here, we develop DeepSignal-plant, a deep learning tool for accurately detecting 5mCs in all three contexts in plants from native Nanopore reads. We have performed Nanopore and bisulfite sequencing of two model plants *Arabidopsis thaliana* (*A. thaliana*) and *Oryza sativa* (*O. sativa*) in parallel. Because cytosines with 100% methylation frequency (fully methylated) are usually much less than cytosines with zero methylation frequency (fully unmethylated) in plants, especially for CHH, it is difficult to collect enough positive training samples from Nanopore reads, which results in an unbalanced training dataset. Therefore, we develop a sample selection strategy to balance and denoise training samples, which can significantly improve the performances of the trained models, especially for CHH and CHG methylation detection. We train one deep learning model in DeepSignal-plant to detect 5mC sites in all three CpG, CHG, and CHH sequence contexts. Testing DeepSignal-plant in *A. thaliana* and *O. sativa* shows a high agreement with bisulfite sequencing. We also test DeepSignal-plant using Nanopore reads of *Brassica nigra* (*B. nigra*), which also achieves high correlations with bisulfite sequencing in 5mC detection of all three contexts. Furthermore, since Nanopore sequencing does not have amplification biases and has a much longer read length than bisulfite sequencing, DeepSignal-plant can profile more 5mC sites in plants than bisulfite sequencing, especially in highly repetitive regions.

## Results

**Profiling cytosine DNA methylations in *A. thaliana* and *O. sativa*.** We have sequenced two model plants *A. thaliana* and *O. sativa* using both bisulfite sequencing and Nanopore sequencing (Methods). For *A. thaliana*, we have sequenced three technical replicates using bisulfite sequencing with ~116×, ~131×, and ~116× mean genome coverage of reads, respectively. For *O. sativa*, we have sequenced two biological replicates using bisulfite sequencing with ~78× and ~126× coverage of reads, respectively. In the *A. thaliana* and *O. sativa* genome, there are 42,859,516 and 162,549,211 cytosines, respectively. For *A. thaliana*, 98.3% cytosines have at least 5× coverage in bisulfite sequencing (Supplementary Table 1). We have observed that about 24.3% CpG, 8.6% CHG, and 3.3% CHH in reads are methylated in all three technical replicates of *A. thaliana* (Supplementary Fig. 1). For *O. sativa*, there are 93.3% and 94.0% cytosines having at least 5× coverage in bisulfite sequencing of two biological replicates (Supplementary Table 1), respectively. In the first replicate (sample1), 52.7% CpG, 27.7% CHG, and 4.5% CHH in reads are methylated and 46.8% CpG, 20.3% CHG, and 2.9% CHH in reads are methylated in the other replicate (sample2) (Supplementary Fig. 2).

We further evaluate the numbers of high confidence fully unmethylated and methylated cytosines in *A. thaliana* and *O. sativa*, which are used for training our model. The number of high confidence fully methylated cytosines is much less than those of high confidence fully unmethylated cytosines in both *A. thaliana* and *O. sativa*, especially in CHG and CHH contexts. The ratios are <1:50, <1:1,000, <1:22,000 for CpG, CHG, and CHH sites in *A. thaliana*, and ~1:2, ~1:15, and ~1:2,000 for CpG, CHG, and CHH sites in *O. sativa*, respectively (Supplementary Fig. 3). The unbalance of high confidence fully unmethylated and methylated cytosines in quantities, especially in non-CpG contexts, creates a challenge to select appropriate samples to train the model for 5mC detection from Nanopore reads.

We generate ~600× Nanopore reads of the same *A. thaliana* sample used in bisulfite sequencing. We also generated ~215×

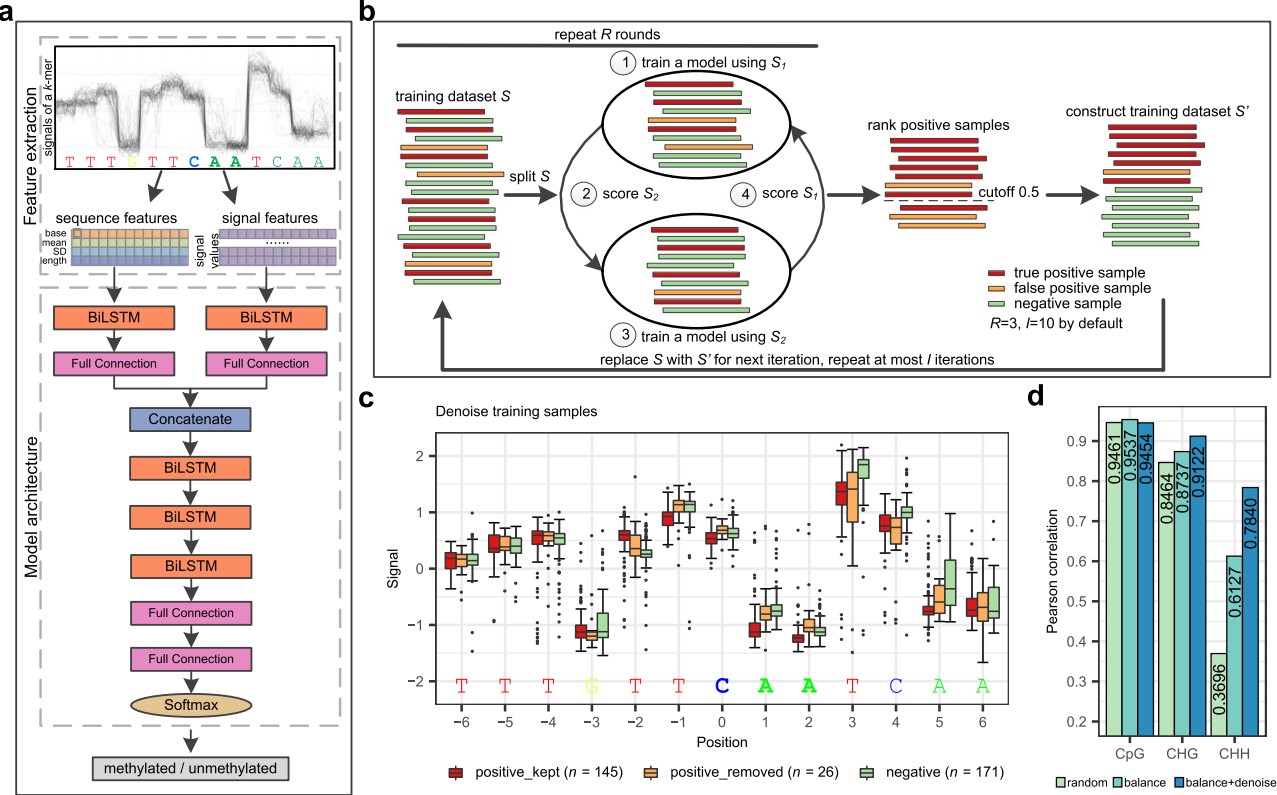

**Fig. 1 DeepSignal-plant for 5mC detection using Nanopore sequencing. a** Architecture of DeepSignal-plant. BiLSTM: a sequence processing network that uses long short-term memory layer to take the input from forward and backward direction to learn order dependence; Full Connection: a fully connected layer that connects all the inputs from the former layer to every activation unit of the next layer; Softmax: an activation function which normalizes a vector of real numbers into a vector of probabilities that sum to 1. **b** Schema of denoise training samples in DeepSignal-plant. **c** Signal comparison of different kinds of samples of a *k*-mer after denoising training. positive_kept: positive samples kept by the *denoising* step; positive_removed: positive samples removed by the *denoising* step; negative: negative samples; *n* = number of signal values for each base; Boxplots indicate 50th percentile (middle line), 25th and 75th percentile (box), the smallest value within 1.5 times interquatile range below 25th percentile and largest value within 1.5 times interquatile range above 75th percentile (whiskers), and outliers (dots). **d** Effectiveness of training samples selection on 5mC detection. The training samples were extracted from ~500× Nanopore reads of *A. thaliana*. Pearson correlations were calculated using the results from ~20× Nanopore reads and three bisulfite replicates of *A. thaliana*.

and ~100× Nanopore reads for the two biological *O. sativa* samples which were used in bisulfite sequencing, respectively.

**The DeepSignal-plant algorithm and training process.** DeepSignal-plant utilizes a bidirectional recurrent neural network[29] (BRNN) with long short-term memory[30] (LSTM) units to detect DNA 5mC methylation from both signal and sequence features of Nanopore reads (Fig. 1a). First, the raw signals of the Nanopore reads are mapped to the nucleotide sequences. Then, for each targeted 5mC site, DeepSignal-plant constructs four *k*-length features, namely the base and the mean, standard deviation, and the number of signal values of each base, of the *k*-mer ($k = 13$ by default) on which the targeted site centers as sequence features. DeepSignal-plant also extracts *m*-length ($m = 16$ by default) signals of each base in the *k*-mer as signal features (Methods). By using BRNN to process both the signal features and sequence features, the size of the DeepSignal-plant model is only one-eighth of the size of DeepSignal[26] (Supplementary Table 2).

We select the high confidence methylated and unmethylated cytosines as training samples based on the bisulfite sequencing results. We select the cytosines with zero methylation frequencies and at least five mapped reads as high confidence unmethylated sites. To include more high-confidence methylated sites, we chose

the cytosines having at least 0.9 methylation frequency and at least five mapped reads (Supplementary Fig. 3, Supplementary Table 3). We preprocess the Nanopore raw reads first by transforming them to a sequence of bases using Guppy and then mapping raw electrical signal values to contiguous bases in genome reference using Tombo[21] (Methods, Supplementary Fig. 4a). We randomly select Nanopore subreads that are aligned to the high confidence 5mC sites for training.

Since the *k*-mers of the selected methylated and unmethylated cytosines are different, especially for the CHH motif, we need to balance *k*-mers in methylated and unmethylated cytosines to avoid the model using the *k*-mer difference to distinguish the 5mC methylation status. Furthermore, for the same *k*-mers, the number of reads for the methylated and unmethylated cytosines are also different. We further balance the numbers of methylated and unmethylated cytosines to train a model for higher performance.

We have selected cytosines whose methylation frequencies are only greater than 0.9, which may introduce false methylated cytosines in the training reads. Therefore, we develop an iterative denoising method to remove false methylated samples (Fig. 1b). In each iteration, we perform a two-fold cross prediction of the training dataset. We remove methylated samples that are predicted as unmethylated by DeepSignal-plant. Then, we use the remaining methylated samples and the unmethylated samples

for the next iteration of training. The denoising method will stop either after 10 iterations, or if there are less than 1% of methylated samples predicted as unmethylated. Figure 1c shows that the electrical signals of the bases in the removed samples of a CHH *k*-mer by the denoising step are similar to those of the unmethylated samples, which demonstrates that the denoising is capable of removing false-positive samples. The denoising can ensure the reliability of methylated samples.

**Evaluation of balancing and denoising methods in DeepSignal-plant**. We first evaluate the denoising method with a simulation experiment (Supplementary Note 1). We generate training datasets with different amounts of mislabeled ratios (i.e., the ratio of false-positive samples to total positive samples, 0–20%) and then process them using our denoising method. The results show that >93% of mislabeled samples are denoised (Supplementary Fig. 5).

We then use bisulfite sequencing as the benchmark and evaluate DeepSignal-plant by calculating the Pearson correlation between the per-site methylation frequencies identified by bisulfite sequencing and Nanopore sequencing. We test the effectiveness of the balancing and denoising methods. We randomly select ~500× Nanopore reads of *A. thaliana* for training, and randomly select ~20× reads (i.e., ~10× reads for forward and the complementary strand of the genome, respectively) from the remaining ~100× coverage of reads for testing. After the denoising, 19.1% of training samples of CHG and 29.4% of training samples of CHH were removed. As shown in Fig. 1d, compared with randomly selecting samples, balancing and denoising training samples can significantly improve the performance of 5mC detection, especially for the CHH. After balancing and denoising training samples, the correlation with bisulfite sequencing increases from 0.8464 to 0.9122 for the CHG, from 0.3696 to 0.7840 for the CHH. The results demonstrate that balancing and denoising can significantly improve CHH and CHG detection. However, the performance for the CpG is not improved after denoising training samples. This may be because, for the CpG, we take the intersection of high confidence methylated sites from all bisulfite replicates as the final high confidence methylated sites. Thus, the training dataset of the CpG is more reliable and can be used to train the model without denoising.

**Evaluation of DeepSignal-plant using Nanopore data of *A. thaliana* and *O. sativa***. Besides training a model for each 5mC context, we also combine the training samples of all three contexts to train one model for whole 5mC detection. The whole 5mC model outperforms the three individual models for CpG, CHG, and CHH (Supplementary Fig. 6). This result indicates that the information of the three motifs can improve each others' methylation prediction. Therefore, we trained the whole 5mC models in our downstream evaluation and analysis.

We first perform cross-chromosomal validation of DeepSignal-plant using *A. thaliana* data. We use the reads from chr1-chr4 for training and test on the reads from chr5 (Supplementary Fig. 7a). DeepSignal-plant achieves high Pearson correlations with bisulfite sequencing on the testing reads (Supplementary Fig. 7b). We then perform a cross-species validation of DeepSignal-plant. Like the above experiments, for *A. thaliana*, we select ~500× and ~20× Nanopore reads for training and testing, respectively. For *O. sativa*, we randomly select ~115× Nanopore reads of the first biological replicate (sample1) for training, and ~20× reads from the remaining ~100× reads for testing. We first train models of DeepSignal-plant using training samples from *A. thaliana* and *O. sativa* Nanopore reads independently and test the models on both *A. thaliana* and *O. sativa* Nanopore reads. Then, we train models

of DeepSignal-plant by combining the training reads of *A. thaliana* and *O. sativa*. As shown in Supplementary Fig. 8, the models trained using the combined reads achieve the overall best performances. For CpG and CHG, both the models trained using reads of individual species and the models trained using the combined reads achieve high correlations with bisulfite sequencing on tested data of both species. For CHH, the model trained by using reads of *A. thaliana* does not perform well on the tested data of *O. sativa*. This may be due to the relatively less amount of high-confidence methylated sites and *k*-mers of CHH in *A. thaliana* than those in *O. sativa* (Supplementary Tables 3, 4). Meanwhile, the model trained using reads of *O. sativa* only achieves similar performance for CHH as the model trained using the combined reads. We further randomly select ~20× reads of *A. thaliana* and *O. sativa* five times and apply the models trained with combined reads to call methylation. We achieve consistent correlations with bisulfite sequencing (standard deviation <0.003) for the five repeated tests. Furthermore, the results of the five tests are also highly correlated with each other (Supplementary Fig. 9), which shows that the predictions of our models are highly reproducible.

**Comparison of DeepSiganl-plant to other tools for 5mC detection**. We compare DeepSignal-plant with Megalodon[28], which can also detect 5mC in all contexts. For a fair comparison, we re-train Megalodon using the same dataset for training DeepSignal-plant. We use ~20× Nanopore reads of *A. thaliana*, *O. sativa* (sample1), and *B. nigra*[31] for the evaluation. As shown in Fig. 2, DeepSignal-plant outperforms both original and re-trained Megalodon in 5mC detection in *A. thaliana* and *O. sativa*, especially for CHH. In *B. nigra*, DeepSignal-plant outperforms both original and re-trained Megalodon in CpG and CHH methylation detection. For CHG detection in *B. nigra*, DeepSignal-plant has a similar result as the re-trained Megalodon while both significantly outperform the original Megalodon.

We then evaluate DeepSignal-plant and the re-trained Megalodon for 5mC detection under different coverage of reads of those three species. DeepSignal-plant outperforms re-trained Megalodon in CpG and CHH methylation detection at all coverages while re-trained Megalodon gets comparable performances with DeepSignal-plant in CHG methylation detection of *O. sativa* and *B. nigra* (Supplementary Fig. 10). We plot the frequency distribution predicted by DeepSignal-plant and re-trained Megalodon using the ~100× Nanopore reads of *A. thaliana* and two samples of *O. sativa* (Supplementary Figs. 11, 12). The plots show that DeepSignal gives a better prediction for methylated CHH than re-trained Megalodon does. Furthermore, DeepSignal-plant achieves lower root mean square errors (RMSE) than re-trained Megalodon does for detecting 5mCs in those three species, except in detecting CHG methylation of *O. sativa* (sample1) (0.0938 vs 0.0934) and *B. nigra* (0.1355 vs. 0.1354) (Supplementary Tables 5, 6). We also evaluate DeepSignal-plant and re-trained Megalodon at read level (Methods, Supplementary Table 7). The results show that DeepSignal-plant gets higher sensitivities than those re-trained Megalodon gets for all motifs of all species. DeepSignal-plant also gets higher accuracies than those of re-trained Megalodon gets, except for the CpG of *B. nigra* (0.9257 vs. 0.9394).

To further assess DeepSignal-plant, we categorize cytosines into three bins based on their methylation frequencies calculated from bisulfite sequencing: low frequency (0.0–0.3), intermediate frequency (0.3–0.7), and high frequency (0.7–1.0) (Methods). Then, we compare the spread of predictions from DeepSignal-plant and re-trained Megalodon (Supplementary Figs. 13–15). Both DeepSignal-plant and re-trained Megalodon have a high

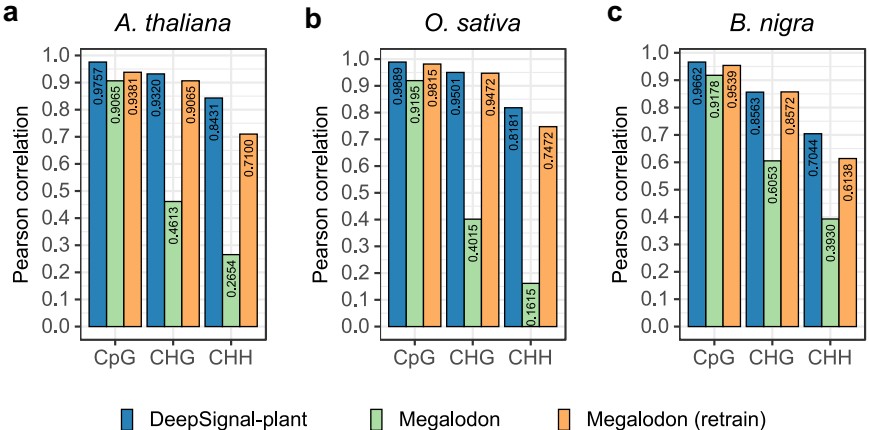

**Fig. 2 Comparison between DeepSignal-plant and Megalodon against bisulfite sequencing on 5mC detection. a** *A. thaliana*, **b** *O. sativa* (sample1), and **c** *B. nigra*. Models of Megalodon (retrain) and DeepSignal-plant were trained using combined reads of *A. thaliana* and *O. sativa*. Pearson correlations were calculated using the results from ~20× Nanopore reads of *A. thaliana*, *O. sativa* (sample1), and *B. nigra* with the corresponding bisulfite replicates, respectively.

consistency with bisulfite sequencing for predicting cytosines with low methylation frequencies in all three contexts. Re-trained Megalodon tends to underpredict cytosines with high and intermediate methylation frequencies in CHH contexts, while the results of DeepSignal-plant are more consistent with those of bisulfite sequencing.

**Nanopore sequencing profiles methylation of more cytosines than bisulfite sequencing.** Previously, we showed that the long reads of Nanopore sequencing can detect more CpG sites than bisulfite sequencing does[26]. Here, we evaluate the 5mCs in all three contexts. With more than 40× coverage of Nanopore reads, DeepSignal-plant can detect more 5mCs than bisulfite sequencing does (Supplementary Fig. 16), which can help profile previously unmappable regions in the genome (Fig. 3a, b, Supplementary Fig. 17) and then make more genome regions be fully profiled (Supplementary Fig. 18). Compared to bisulfite sequencing, DeepSignal-plant detects 1.1% more 5mCs in *A. thaliana* and 5.3% more 5mCs in *O. sativa* with 100× coverage (Supplementary Table 1, Supplementary Figs. 19–20). Especially, DeepSignal-plant detects 1.4% more CHGs and 1.5% more CHHs in *A. thaliana* and at least 5.1% more CHGs and 5.8% more CHHs in *O. sativa*. The *k*-mers around cytosines detected only by DeepSignal-plant have a significant overlap with the *k*-mers used for training DeepSignal-plant (Supplementary Table 8), which indicates *k*-mers around cytosines in repeat regions are similar to those in mappable regions. While the CpGs detected by DeepSignal-plant only are either with low or with high methylation frequency, most of CHHs detected by DeepSignal-plant only are with low methylation frequency. CHGs detected by DeepSignal-plant only in *A. thaliana* tend to have low methylation frequency, while CHGs detected by DeepSignal-plant only in *O. sativa* have either with low or with high methylation frequency (Supplementary Fig. 21).

It is not surprising that a significant amount of cytosines profiled by DeepSignal-plant only exists in centromeres, pericentromeric and telomeres areas, which are composed of thousands of repeats[32] (Fig. 3c, d, Supplementary Figs. 22, 23). Furthermore, many of those newly profiled cytosines are in protein-coding genes and transposons of *A. thaliana* and protein-coding genes of *O. sativa*. DeepSignal-plant has newly profiled the methylation status of 341 genes in *A. thaliana* (Supplementary Data 1) and 227 genes in *O. sativa* (Supplementary Data 2–4). Moreover,

inside the gene body, those cytosines are dominated in the CDS region for both *A. thaliana* and *O. sativa* (Supplementary Fig. 23).

**Differentially methylated cytosines in repeat pairs.** Identification of the methylation status of cytosines in repetitive genome regions is important for understanding gene-regulating and repeat-associated disorders[33]. Here, we evaluate the methylation status of cytosines located within segmental duplications in *A. thaliana* and *O. sativa*. We first generate repeat pairs in the genome of *A. thaliana* and *O. sativa* using MUMmer[34] (Methods). We treat two regions that have length >=100 and the alignment identity >=0.99 as a repeat pair. We count the number of differentially methylated cytosines in each repeat pair (Methods). Then, we define a repeat pair as differentially methylated if there are at least 10% cytosines (or CGs, CHGs, CHHs) that are differentially methylated between them (Methods). We find that over ~9 and ~6% repeat pairs in *A. thaliana* and *O. sativa* are differentially methylated, respectively (Fig. 4a, b, Supplementary Fig. 24a, Supplementary Table 7). Furthermore, we find that the motifs of differentially methylated cytosines in repeat pairs are species-specific (Fig. 4c, d, Supplementary Fig. 24b, 25): CpG sites are more likely differentially methylated in repeat pairs of *A. thaliana*, while CHG sites are more likely differentially methylated in *O. sativa*. Compared to bisulfite sequencing, DeepSignal-plant identifies more differentially methylated repeat pairs (Supplementary Fig. 26). There are several >1000 long repeat pairs in *A. thaliana* and >10,000 long repeat pairs in *O. sativa*, which are differentially methylated (Fig. 4f, g, Supplementary Fig. 27, Supplementary Table 9). The differentially methylated repeat pairs in two replicates of *O. sativa* show a great consistency, which implies that the differentially methylated repeat pairs are stable in species (Fig. 4e, Supplementary Fig. 28). The methylation status of 5mC in the repetitive genome region may provide insights into the relationship between duplicate gene transcription and methylation signatures[35]. Furthermore, as paralogous sequence variants can be used to resolve segmental duplications[36], the differential methylation between repeat pairs may be helpful to resolve collapsed regions of segmental duplications in de novo assemblies of plants.

## Discussion

In this study, we propose a deep learning tool, DeepSignal-plant, to detect DNA 5mCs in plants from native Nanopore reads. With

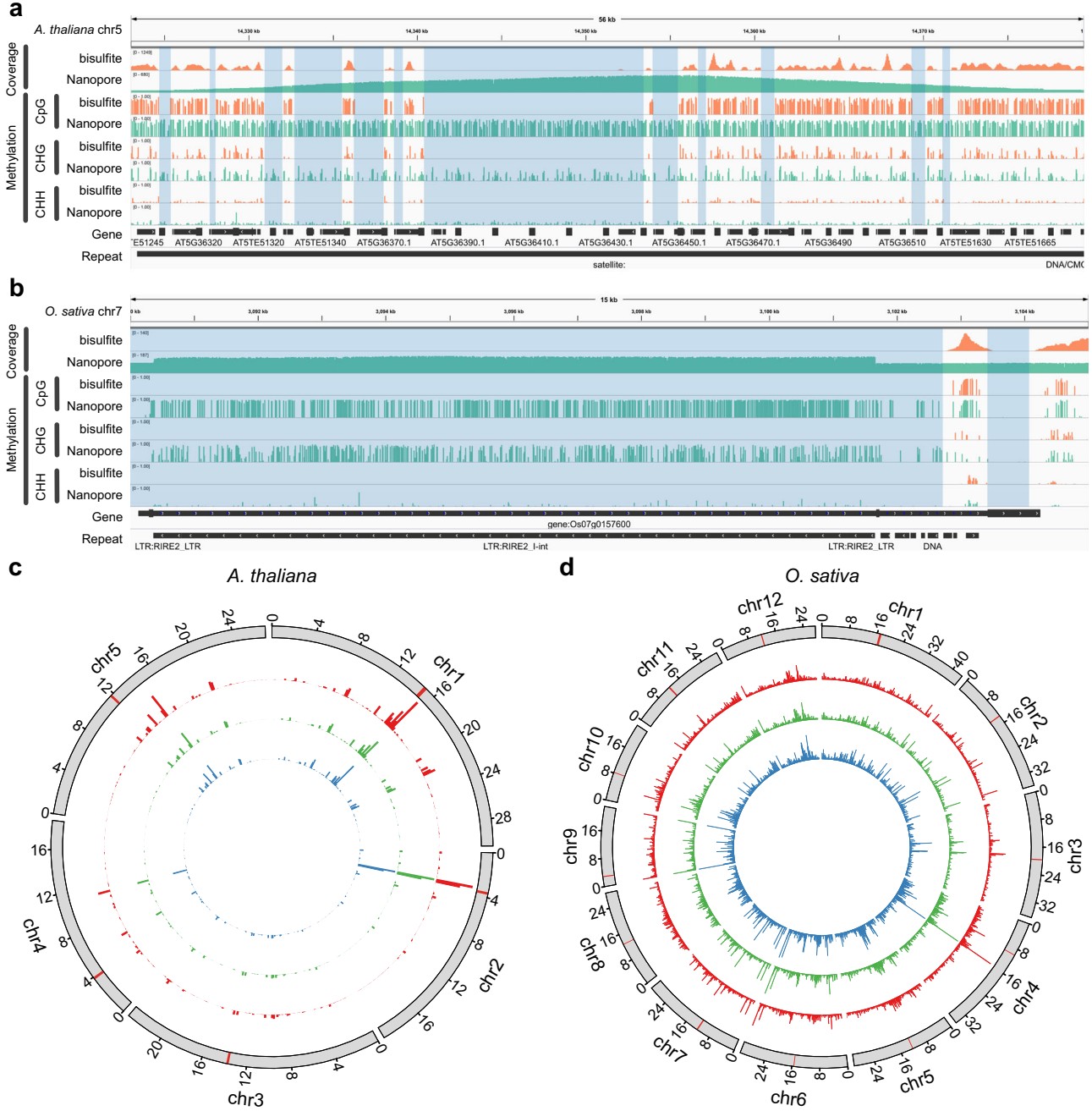

**Fig. 3 DeepSignal-plant profiles methylation status of more cytosines than bisulfite sequencing does. a**, **b** Genome browser view[53] of the reads coverage and methylation status of a 56 kb region (chr5:14323000–14379500:+) of *A. thaliana* (**a**) and a 15 kb region (chr7:3089990–3104990:+) of *O. sativa* (sample1) (**b**). The blue shaded areas show the gaps which cannot be mapped by bisulfite sequencing. **c**, **d** Circos plot[54] of the number of cytosines detected by Nanopore sequencing only in the genomes of *A. thaliana* (**c**) and *O. sativa* (sample1) (**d**). Cycles from inner to outer: CpG (blue), CHG (green), CHH (red), reference (the chromosomes are binned into 200,000-bp (base pair) windows. The centromeric region is indicated by the red bar in each chromosome). Source data are provided as a Source Data file.

a denoising training sample method, DeepSignal-plant models are trained to accurately detect 5mCs in all three contexts. Experiments on Nanopore data of three plants (*A. thaliana*, *O. sativa*, and *B. nigra*) show that DeepSignal-plant has a high agreement with bisulfite sequencing for the predictions of CpG, CHG, and CHH methylation. DeepSignal-plant can detect 5mC sites with acceptable accuracies even with low coverage of reads. Furthermore, DeepSignal-plant can profile methylation of more cytosines in plants than bisulfite sequencing does, especially in highly

repetitive genome regions, which will have more advantages for repetitive plant and polyploidy plant genomes. For example, DeepSignal-plant can identify the 5mC methylation status of a cluster of 61 tRNA genes that have not been detected by the bisulfite sequencing in a previous study[37]. In another study about the DNA methylation patterns of the NB-LRR-encoding gene family in *A. thaliana*[38], three NB-LRR-encoding genes, *At1g58807*, *At1g59124*, *At1g59218*, whose 5mC methylation status can be detected by DeepSignal-plant, was not included. We reanalyze the

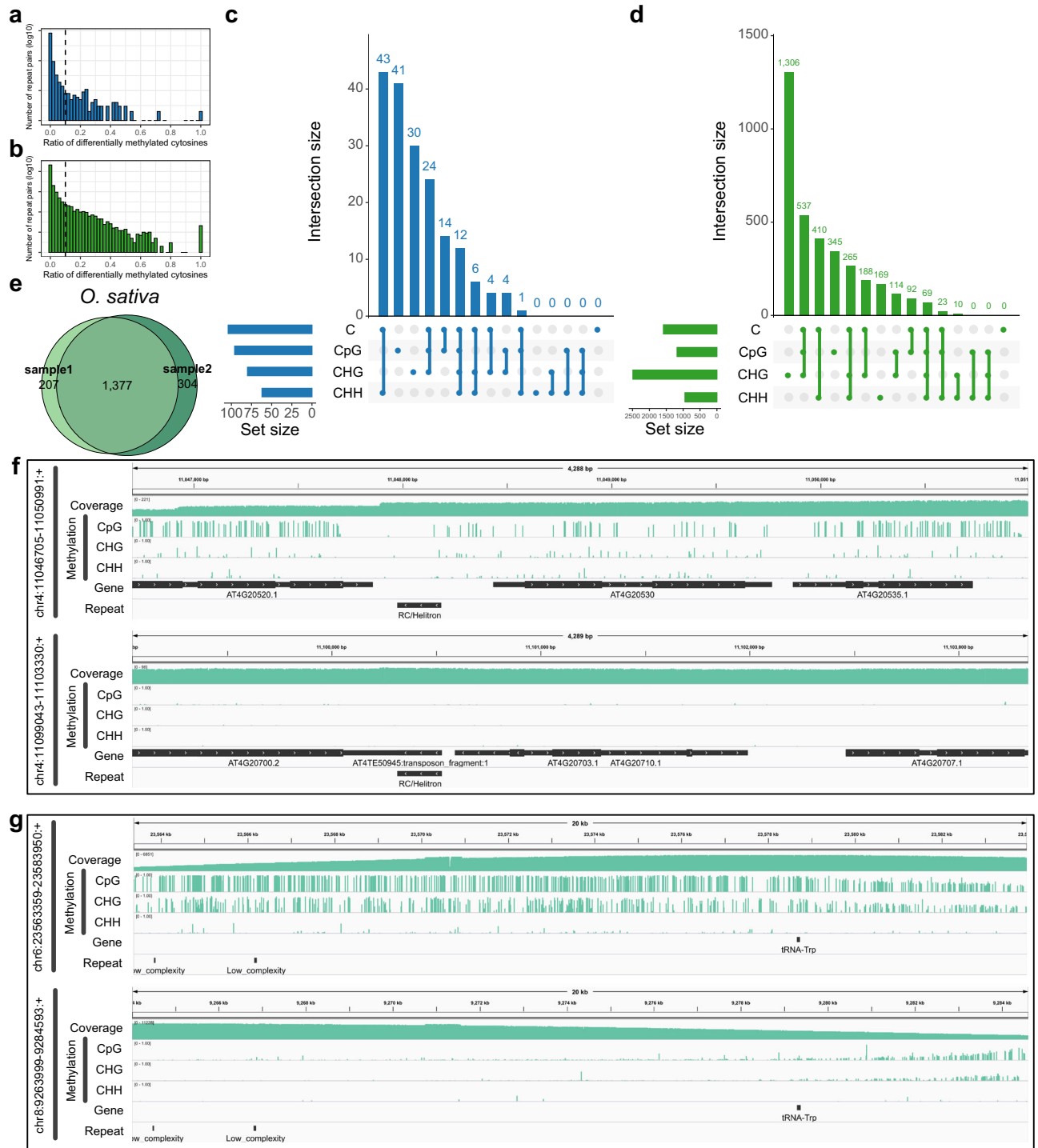

**Fig. 4 Differentially methylated repeat pairs identified by DeepSignal-plant. a**, **b** Ratio of differentially methylated cytosines to total cytosines in each repeat pair of *A. thaliana* (**a**) and *O. sativa* (sample1) (**b**). The black dash lines (10%) indicate repeat pairs are differentially methylated (right) or not (left). **c**, **d** Matrix layout[55] for all intersections of four sets of differentially methylated repeat pairs profiled by cytosines, CpG sites, CHG sites, and CHH sites independently, in *A. thaliana* (**c**) and *O. sativa* (sample1) (**d**). Circles below in each column indicate sets that are part of the intersection; the up bars indicate the size of each intersection; the left bars indicate the total size of each set. **e** Comparison of differentially methylated repeat pairs identified by methylation of cytosines in *O. sativa* sample1 and sample2. **f** Genome browser view of a differentially methylated repeat pair (chr4:11046705–11050991:+, chr4: 11099043–11103330:+) in *A. thaliana*. **g** Genome browser view of a differentially methylated repeat pair (chr6:23563359–23583950:+, chr8: 9263999–9284593:+) in *O. sativa* (sample1). Source data underlying **f** and **g** are provided as a Source Data file.

bisulfite sequencing data and find that there is no read covering those three genes. DeepSignal-plant can become a well-applicable method for 5mC detection in plants, which will provide novel and deeper insights into the epigenetic mechanisms of plants.

## Methods

**Plant materials and DNA extraction**. Wild type Arabidopsis thaliana (L.) Heynh. Columbia-0 (Col-0) was used in this study. Seeds were surface sterilized in 4% sodium hypochlorite, vernalized for 2 days at 4 °C, and grown on half Murashige & Skoog (MS) plates for 7days at 22 °C, 70% relative humidity with a 16 h/8 h

light/dark regime. Seven-day-old seedlings were then transplanted into individual pots with soil and cultivated for one month in the same condition as seedlings on half MS plates. For *O. sativa*, we used wild-type Oryza sativa L. ssp. Japonica cv. Nipponbare. Seeds were soaked in distilled water for 48 h at 37 °C to accelerate germination. Then, the germinated seeds were sown in soil and cultured for 1 month in a growth chamber at 28 °C, 70% relative humidity with a 14 h/10 h light/dark regime. As for taking samples of *O. sativa*, each seedling was divided into two halves in the vertical direction with scissors, marked with "left group" and "right group." For *A. thaliana* and *O. sativa*, the harvested material was from at least 30 individual plants and quickly froze with liquid nitrogen, stored at −80 °C until further use. Genomic DNA was extracted from samples by QIAGEN® Genomic DNA extraction kit (Cat#13323) according to the manufacturer's standard operating procedure. The extracted DNA was detected by NanoDrop™ One UV-Vis spectrophotometer (Thermo Fisher Scientific) for DNA purity. Then Qubit® 3.0 Fluorometer (Invitrogen) was used to quantify DNA accurately.

**Bisulfite sequencing**. The extracted genomic DNA was first sheared by Covaris and purified to 200–350 bp in average size. Sheared DNA was then end-repaired and ligated to methylated sequencing adapters. Finally, adapter-ligated DNA was bisulfite-converted and PCR-amplified. The bisulfite conversion kit and library preparation kit for sequencing three technical replicates of *A. thaliana* and *O. sativa* (sample2) were TIANGEN DNA Bisulfite Conversion Kit (cat #: DP215, TIANGEN BIOTECH) and TruSeq DNA Methylation Kit (cat #: EGMK91324, Illumina), respectively. For *O. sativa* (sample1), the bisulfite conversion kit and library preparation kit were EZ DNA Methylation-Gold Kit (Zymo Research) and MGIEasy Whole Genome Bisulfite Sequencing Library Prep Kit (16 RXN) (BGI), respectively. The libraries of three technical replicates of *A. thaliana* were sequenced on a NovaSeq6000 sequencer (Illumina) to obtain pair-end 150 bp (base pair) reads. ~116×, ~131×, and ~116× mean genome coverage of reads for each replicate were generated. For *O. sativa*, two biological replicates were sequenced: the library of one biological replicate (sample1) was sequenced on an MGI2000 (BGI) sequencer to obtain pair-end 100 bp reads (~78× coverage of reads); the library of the other replicate (sample2) was sequenced on a NovaSeq6000 sequencer (~126×). The sequencing reads were then processed by the standard pipeline of Bismark[39] (v0.20.0). For each detected cytosine in CpG, CHG, and CHH motif, Bismark outputs a methylation call for each of its mapped reads. Then, the methylation frequency of the cytosine is calculated, which is the number of mapped reads predicted as methylated divided by the number of total mapped reads.

**Nanopore sequencing**. The extracted DNA was qualified, size-selected using the BluePippin system (Sage Science). Then, the genomic DNA was end-repaired and PCR adapters supplied in the Oxford Nanopore Technologies (ONT) sequencing kit (SQK-LSK109) were ligated to the end-repaired DNA. Finally, Qubit® 3.0 Fluorometer (Invitrogen) was used to quantify the size of library fragments. To generate Nanopore reads, the prepared libraries are loaded into flow cells (R9.4, FLO-PRO002) of a PromethION sequencer (ONT). Raw Nanopore reads were then basecalled by Guppy (version 3.6.1 + 249406c), the official basecaller of ONT, with *dna_r9.4.1_450bps_hac_prom.cfg*. In total, there were 3,124,608 reads with an average length of 23,751 bp (~600×) for *A. thaliana*. For *O. sativa*, reads for each of the two biological replicates were generated: 3,274,036 reads with an average length of 25,990 bp (~215×) for sample1 and 1,671,237 reads with an average length of 23,790 bp (~100×) for sample2, respectively.

**Data partition of Nanopore reads of *A. thaliana* and *O. sativa***. For *A. thaliana*, we randomly selected ~500× Nanopore reads for training. The remaining ~100× reads were used for evaluation. For *O. sativa*, ~115× reads of one replicate (sample1) were randomly selected for training. The remaining ~100× reads, together with all ~100× Nanopore reads of the other replicate (sample2) were used for evaluation (Supplementary Table 10).

**Genome references and annotations**. The genome reference of *A. thaliana* was downloaded from NCBI with the version GCF_000001735.4_TAIR10.1[40]. The gene annotation and centromere location of *A. thaliana* were downloaded from Araport11[41]. The genome reference and gene annotation of *O. sativa* were downloaded from EnsemblPlants with the version IRGSP-1.0[42] (Assembly GCA_001433935.1). Locations of centromeres in *O. sativa* were downloaded from Rice Annotation Project Database[43]. Repeat regions of *A. thaliana* and *O. sativa* were downloaded from NCBI Genome Data Viewer with the 'RepeatMasker' track in the corresponding genomes[40,44]. The tandem repeats and inverted repeats were generated by Tandem Repeats Finder[45] (version 4.09) and Inverted Repeats Finder[46] (version 3.05) with corresponding genome references and suggested parameters, respectively.

**Select high-confidence sites from bisulfite sequencing**. We used bisulfite sequencing as the gold standard to train DeepSignal-plant models for 5mC detection from Nanopore reads. From the results of bisulfite sequencing, we took cytosines covered with at least five reads and had at least 0.9 methylation frequency as high-confidence methylated sites. Cytosines that had at least five mapped reads and zero methylation frequency were selected as high-confidence unmethylated sites. For the CpG motif of *A. thaliana*, we took the intersection of high-confidence

sites from all three technical replicates as the final high-confidence sites set to train models. For CHG and CHH motif, we took the sites whose methylation frequencies are zero in all bisulfite replicates as the final high-confidence unmethylated sites. The methylation levels of CHG and CHH sites are relatively lower in *A. thaliana*. Thus, we took the union of the high confidence methylated sites from three replicates as the final high-confidence methylated sites of CHG and CHH motif (Supplementary Table 3). For *O. sativa*, high-confidence sites from sample1 were selected in a similar way (Supplementary Table 3).

**The framework of DeepSignal-plant**. DeepSignal-plant takes the raw reads of Nanopore sequencing and a reference genome as input. Before using DeepSignal-plant to call methylation, raw reads must be pre-processed (Fig. 1a) by the following two steps:

1. Basecall. We use Guppy (version 3.6.1 + 249406c) for the basecalling of all the Nanopore reads.
2. Re-squiggle. We use Tombo[21] (version 1.5.1) to map raw signals of reads to contiguous bases in the genome reference. In Tombo, minimap2[47] (version 2.17-r941) is used for the alignment between reads and genome reference. Tombo corrects the insertion and deletion errors in Nanopore reads and re-annotates raw signals to match the genomic bases.

After pre-processing of raw reads, DeepSignal-plant trains models and calls methylation (Supplementary Fig. 4a) as the following steps:

1. Extract features. After re-squiggle, raw signals of each read are normalized by using median shift and median absolute deviation (MAD) scale first[26]. Then, for each base in one read, we can get the set of normalized signals mapped to the base. Therefore, for each targeted site, we can use the $k$-mer where the targeted site centers on it and the corresponding normalized signals to form two groups of feature vectors: (1) *Sequence features*. For each base in the $k$-mer, we calculate the mean, standard deviation, and the number of its mapped signal values. Thus, we construct a $k \times 4$ matrix as sequence features, where there are 4 features for each base of the $k$-mer: the nucleotide base, the mean, standard deviation, and the number of signal values of each base, respectively. (2) *Signal features*. We sample $m$ signals from all signals of each base to form a $k \times m$ matrix as signal features. For each base, if the number of signals is less than $m$, we paddle with zeros. We set $k = 13$ and $m = 16$ as default to extract features of each targeted site.
2. Model architecture. BRNN[29] with LSTM units[30] is used in the model of DeepSignal-plant (Fig. 1a, Supplementary Note 2). An BRNN is a neural network model for sequential data. Each BRNN includes a forward RNN and a backward RNN to catch both the forward and backward context. An RNN scans the sequence of data and encodes the sequential information into a latent representation. In detail, sequence features and signal features, are each fed into a BRNN layer, followed by a fully connected layer. Then, the output features are concatenated and fed into another three-layer BRNN and two fully connected layers. Finally, a softmax activation function is used to output two probabilities $P_m$ and $P_{um}$ ($P_m + P_{um} = 1$), which represent the probabilities of methylated and unmethylated, respectively.
3. Train models. To train a model of DeepSignal-plant, the selected training samples (by the balancing and denoising method) from Nanopore reads are split into two datasets for training and validation at a ratio of 99:1. We use Adam optimizer[48] to learn model parameters on the training dataset by minimizing the loss calculated by cross-entropy (Supplementary Note 3). The model parameters which get the best performance on the validation dataset are saved. To prevent overfitting, we use two strategies. First, we use dropout layers[49] in LSTM layers and fully connected layers. Second, we use early stopping[50] during training. The model parameters with the current best performance on the validation dataset are saved in every epoch. If the best performance of the current epoch decreases, we stop the training process. Hyperparameter tuning of DeepSignal-plant is also performed (Supplementary Note 3). According to the experimental results, the $k$-mer length in DeepSignal-plant is set to 13 (Supplementary Fig. 29). The number of signals used to construct signal features is set to 16 (Supplementary Fig. 30). The number of BiLSTM layers to process the concatenated sequence and signal features, the number of hidden units in each BiLSTM layer, and the initial learning rate for training are set to 3, 256, and 0.001, respectively (Supplementary Fig. 31, Supplementary Table 11). And models using both sequence and signal features to call methylation are shown to have the best performances (Supplementary Fig. 32b).
4. Call methylation and calculate methylation frequency. For a targeted cytosine site in a read, DeepSignal-plant outputs the methylated probability $P_m$ and unmethylated probability $P_{um}$. If $P_m > P_{um}$, the site is called methylated, otherwise is called unmethylated. Then, by counting the number of reads where the site is called methylated and the total number of reads mapped to the site, DeepSignal-plant calculates a methylation frequency of the site.

DeepSignal-plant is implemented in Python3 and PyTorch (version 1.2.0). The evaluation of running time and peak memory usage of the DeepSignal-plant pipeline is shown in Supplementary Note 4 and Supplementary Table 12.

**Balance training samples of each *k*-mer**. During the training of DeepSignal-plant, we first extract training samples from all reads aligned to the high-confidence sites. Then, we randomly subsample at most 20 million (half positive and half negative) samples as the training dataset for each motif. However, there exist big differences between the *k*-mers in the selected high-confidence methylated and unmethylated sites (Supplementary Table 4), which leads to unbalanced positive and negative samples for each *k*-mer in the training dataset. The unbalanced training data can significantly affect model training, especially for the CHH motif. Therefore, we balance the types of *k*-mer in positive and negative samples, as well as the number of reads covered for each *k*-mer. The algorithm **1** for Balance_Negative_Samples ($S_{pos}$, $S_{neg}$) is as follows:

Input: a set of positive samples $S_{pos}$, set of negative samples $S_{neg}$
Output: a set of balanced negative samples $S'_{neg}$

1. $K_{pos}$ = set of *k*-mers in $S_{pos}$, $K_{neg}$ = set of *k*-mers in $S_{neg}$
2. $K_{comm}$ = $K_{neg}$.intersection($K_{pos}$), $K_{diff}$ = $K_{neg}$.difference($K_{pos}$)
3. $KNUM_{pos}$ = number of samples of each *k*-mer in $S_{pos}$
4. $S'_{neg}$ = Ø
5. for each *k*-mer $k$ in $K_{comm}$ do
6. $k\_count$ = $KNUM_{pos}$ ($k$)
7. $S'_{neg\_k}$ = set of at most $k\_count$ samples of $k$ extracted from $S_{neg}$ randomly
8. $S'_{neg}$ += $S'_{neg\_k}$
9. if $|S_{pos}| - |S'_{neg}| > 0$ then
10. $S'_{neg}$ += set of $|S_{pos}| - |S'_{neg}|$ samples of *k*-mers in $K_{diff}$ extracted from $S_{neg}$ randomly
11. return $S'_{neg}$

**Denoise training samples**. There exist false-positive samples in the training dataset. We design an algorithm that can iteratively remove false-positive samples in the training dataset (Fig. 1b). In this denoising algorithm, we balance the samples using Algorithm 1 after each iteration of the denoising procedure. The algorithm **2** for Denoise_Samples ($S$, $I$, $R$, $E$) is as follows:

Input: training samples set $S$, number of iterations $I$, number of rounds $R$, number of epochs $E$
Output: denoised training samples set $S'$

1. $S_{neg}$ = set of negative samples in $S$
2. for $i$ in 1: $I$ do
3. initialize *score*, assign $score(s) = []$ for each sample $s$ in $S$
4. for $r$ in 1: $R$ do
5. randomly split $S$ into two equal sets $S_1$ and $S_2$; initialize $score\_r$
6. train a DeepSignal-plant model *model1* for $E$ epochs using $S_1$
7. $score\_r(s)$ = the methylation probability of $s$ predicted by *model1* for each sample $s$ in $S_2$
8. train a DeepSignal-plant model *model2* for $E$ epochs using $S_2$
9. $score\_r(s)$ = the methylation probability of $s$ predicted by *model2* for each sample $s$ in $S_1$
10. $score(s)$.append($score\_r(s)$) for each sample $s$ in $S$
11. $S_{pos}$ = set of positive samples in $S$, $P\_total = |S_{pos}|$
12. for each sample $s$ in $S_{pos}$ do
13. if mean($score(s)$) < 0.5 then
14. $S_{pos}$.remove($s$)
15. $S'_{neg}$ = Balance_Negative_Samples ($S_{pos}$, $S_{neg}$)
16. $S = S_{pos} + S'_{neg}$
17. if $|S_{pos}|/P\_total >= 0.99$ then
18. break
19. $S' = S$
20. return $S'$

In the denoising algorithm, we set $I = 10$, $R = 3$, $E = 3$ as default. According to our experiments, using only signal features in DeepSignal-plant to denoise training samples got the best performance (Supplementary Fig. 32a).

**Evaluation of the proposed pipeline on *A. thaliana* and *O. sativa***. We use bisulfite sequencing as the benchmark to test the trained models of DeepSignal-plant. To compare with bisulfite sequencing, we use cytosines from both forward and complementary strands of genomes of *A. thaliana* and *O. sativa*. 5 chromosomes of *A. thaliana* and 12 chromosomes of *O. sativa* are used. (1) *Comparison of the number of cytosines detected by bisulfite and Nanopore sequencing.* For comparison, we count sites that have at least five mapped reads in both Nanopore sequencing and bisulfite sequencing, respectively. In bisulfite sequencing of *A. thaliana*, we count sites that have at least 5 mapped reads in at least one technical replicate. (2) *Comparison of methylation frequencies.* We use Pearson correlation ($r$), together with the coefficient of determination ($r^2$), Spearman correlation ($\rho$), and root mean square error (RMSE), to compare per-site methylation frequencies calculated by Nanopore sequencing and the corresponding replicates of bisulfite sequencing. Cytosines with at least five mapped reads in both bisulfite and Nanopore sequencing are selected for evaluation. For *A. thaliana*, we calculate the average correlations between Nanopore sequencing and three bisulfite replicates. To calculate methylation frequencies with different coverage of Nanopore reads, we randomly select reads from all testing reads of *A. thaliana* and *O. sativa*. (3)

*Comparison of lowly, intermediately, and highly methylated sites.* A site is said to be lowly methylated if it has at least five mapped reads and the methylation frequency of the site is at most 0.3. A site is highly methylated if it has at least five mapped reads and the methylation frequency of the site is at least 0.7. The cytosines with methylation frequencies between 0.3 and 0.7 and at least five mapped reads are categorized as intermediately methylated sites. For Nanopore sequencing, we categorized the cytosines based on the methylation frequencies predicted from the ~100× reads selected. For bisulfite sequencing of *A. thaliana*, we count a site as lowly, intermediately, or highly methylated if the site is lowly, intermediately, or highly methylated in all three replicates. (4) *Evaluation at read level.* To evaluate at read level, we first select cytosines with one and zero methylation frequency based on bisulfite sequencing. Then we extract corresponding positive and negative samples of the selected sites from Nanopore reads. We randomly select 100,000 positive samples and 100,000 negative samples for evaluation. After the subsampling, accuracy, sensitivity, specificity, and area under the curve (AUC) are calculated using Python3 and scikit-learn (version 0.20.1) package. The sub-sampling evaluation is repeated five times.

**Evaluation of the proposed pipeline on *B. nigra***. We got ~78× Nanopore reads and the de novo assembly (Bnigra_NI100.v2.genome.fasta, 491 M, 8 chromosomes) of *B. nigra* Ni100 from Parkin et al.[31]. The cytosine methylation profile from bisulfite sequencing was also provided by Parkin *et al.*, which was generated by BSMAP[51] (v2.9) from ~20× bisulfite sequencing reads. Cytosines from both the forward and complementary strands of eight chromosomes of *B. nigra* were included for evaluation. According to bisulfite sequencing, there are 70.2% CpG, 27.9% CHG, and 8.3% CHH methylation at read level in *B. nigra*. To compare methylation frequencies with bisulfite sequencing, cytosines that have at least 5 mapped reads in Nanopore sequencing were selected. Since the coverage of bisulfite sequencing is insufficient, we selected cytosines with at least 10 mapped reads in bisulfite sequencing for evaluation.

**Retrain megalodon**. Two model configuration files of Megalodon[28], res_dna_r941_prom_modbases_5mC_CpG_v001.cfg and res_dna_r941_min_modbases-all-context_v001.cfg, (version 2.2.3), were used to detect 5mCs in CpG and non-CpG contexts, respectively. To train a new model of Megalodon, an initial model is needed. Using the configuration file res_dna_r941_min_modbases-all-context_v001.cfg as the initial model, we trained a new 5mC model of Megalodon by training 2 rounds with the selected Nanopore reads of *A. thaliana* and *O. sativa*, respectively. A detailed description of training models of Megalodon is shown in Supplementary Note 5 and Supplementary Fig. 4b.

**Identify repeat pairs**. To identify repeat pairs (i.e., two same/similar sequences in genome reference), we first used MUMmer[34] (version 4.0.0beta2) to align the genome reference to itself. Then, from the results of MUMmer, we used in-house Python scripts to select two regions of which length >100 and identity score >0.99 as repeat pairs. Finally, we kept repeat pairs that contain at least one cytosine for analysis. Suppose the methylation frequencies of a cytosine in the same relative position of the repeat pair are $rmet_1$ and $rmet_2$, the cytosine is said to be differentially methylated if $|rmet_1 - rmet_2| >= 0.5$. A repeat pair is said to be differentially methylated if there are at least 10% cytosines (or CpG sites, CHG sites, CHH sites independently) that are differentially methylated in the repeat pair.

**Reporting summary**. Further information on research design is available in the Nature Research Reporting Summary linked to this article.

## Data availability

Data supporting the findings of this work are available within the paper and its Supplementary Information files. A reporting summary for this article is available as a Supplementary Information file. All sequencing data generated in this study (bisulfite sequencing and Nanopore sequencing data of *A. thaliana* and *O. sativa*) have been deposited in the National Center for Biotechnology Information (NCBI) under BioProjectID PRJNA764549 and Sequence Read Archive (SRA) accession No. SRP337810, as well as in the Genome Sequence Archive of BIG Data Center, Beijing Institute of Genomics (BIG, http://gsa.big.ac.cn), Chinese Academy of Sciences, with Project accession No. PRJCA004326 and GSA accession No. CRA003885. Nanopore and bisulfite sequencing data of *B. nigra*[31] are available at NCBI BioProject ID PRJNA516907. The gene annotation of *A. thaliana*[41] is available at TAIR [https://www.arabidopsis.org/index.jsp]. The gene annotation of *O. sativa*[42,43] is available at EnsemblPlants [https://plants.ensembl.org/Oryza_sativa/Info/Index] and RAP-DB [https://rapdb.dna.affrc.go.jp/index.html]. Genes that cannot be covered by bisulfite sequencing, but can be covered by Nanopore sequencing in *A. thaliana* and *O. sativa* are provided in Supplementary Data 1–4. Source data are provided with this paper.

## Code availability

DeepSignal-plant and a detailed tutorial are publicly available at GitHub [https://github.com/PengNi/deepsignal-plant] and Zenodo[52]. Code for reproducing results and analysis was documented at GitHub [https://github.com/PengNi/plant_5mC_analysis].

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

## Acknowledgements

We thank Prof. Parkin and her colleagues for sharing their bisulfite sequencing and Nanopore sequencing data of *B. nigra*. This work was supported in part by the National Natural Science Foundation of China under Grants (Nos. U1909208, 61732009 and 61832019), 111 Project (No. B18059), Hunan Provincial Science and Technology Program (No. 2018wk4001) to J.W., the National Natural Science Foundation of China under Grant (No. 91953122) to C.L.X., the U.S. National Institute of Food and Agriculture (NIFA) under Grant 2017-70016-26051 and the U.S. National Science Foundation (NSF) under Grant ABI-1759856 and MTM2-2025541 to F.L.

## Author contributions

J.X.W., F.L., and C.L.X. conceived and designed this project. P.N., J.X.W., and F.L. conceived, designed, and implemented the pipeline and the model of DeepSignal-plant. W.L. and L.B. provided plant materials and DNA samples of *A. thaliana* and *O. sativa*. C.L.X. helped sequencing the Nanopore and bisulfite data of *A. thaliana* and *O. sativa*.

P.N., N.H., F.N., J.Z., and Z.Z. evaluated DeepSingal-plant using the sequenced data. P.N. and B.W. annotated the methylation results in the genome of *A. thaliana* and *O. sativa*. P.N., F.L., J.X.W., and C.L.X. wrote the paper. All authors have read and approved the final version of this paper.

## Competing interests

The authors declare no competing interests.
