## [Peer Review File · Nature Communications]

Genome-wide Detection of Cytosine Methylations in Plant from Nanopore Data Using Deep LearningREVIEWER COMMENTS

Reviewer #1 (Remarks to the Author):

The authors describe a method to detect Cytosine methylation in the context of plants, where cytosine methylation occurs not only in CpG sites, but also in CHG and CHH trinucleotides. Existing (methylated) base callers have not been optimized for this context. They authors showcase the method in three plant models, and train and test their caller using bisulphite sequencing data. They show that their method can produce data similar to bisulphite sequencing and detects methylation in sites that are not mappable using short read sequencing.

The detection of DNA modifications in a context unbiased manner is a very relevant research question. Research that helps improve models in this regard are very important and relevant for the Nanopore community.

Major concerns:

- The authors show that the denoising method improves performance by improving the correct labelling of the samples. While this might be true, the approach itself looks like a self-fulfilling prophecy, as the same samples that are re-labelled are going to be used for training. A simulation experiment should be done to show that indeed the denoising method works as intended, this can be easily done by having a ground-truth dataset and purposely mislabelling several samples and evaluating whether the method would correct such mislabellings.
- Nanopore sequencing combined with DeepSignal-plant detects many novel methylated sites in repeat regions. However there is no control to establish whether these calls are accurate or false positives. It could very well be possible that the kmers in repeat regions are significantly different from those in mappable regions, on which the algorithm was trained. The authors could show that there is significant overlap between the kmers in repeat regions, and the kmers used for training. If this overlap is not significant, this issue could for example be addressed by using synthetic controls, to validate that the model calls methylation correctly in previously unmappable regions.
- The strategy employed to detect methylated cytosines uses a segmentation approach. This type of models was used in the past for basecalling, but nowadays state of the art basecallers do not require segmented data for prediction (eg. Chiron, Guppy, Bonito, Megalodon, ...). In this regard, it feels like this strategy is a step back regarding the technological advances of the past years.
 - Because of this, authors are bound to sample raw data datapoints for each base if there are more than their network input size allows.
 - This approach is also bound by the tool Tombo, future changes in the tool might affect performance of the models. This might also require re-training of the models for each Tombo version.
 - Finally, this also forces the need of having a reference genome available.
- The authors compare their model with Megalodon. While it is true that megalodon can be trained to detect base modifications, Megalodon's task is also to do regular basecalling. For this reason, Megalodon's task is far more complex than DeepSignal-plant, as the latest only has to work on methylation detection.
 - It is mentioned that two Megalodon configurations are trained (methods section), but in the end only 3 models are trained. It is unclear which of the two configurations was used in the end.
 - It is unclear if the authors also employed their denoising approach to improve the dataset during Megalodon training.
 - It is unclear if the authors also employed their k-mer balancing approach for Megalodon training.
 - For these reasons, I believe the comparison to be a bit unfair towards Megalodon.

- The authors compare the output of their model to bisulphite sequencing and measure its performance using pearsson correlation. However, sensitivity and accuracy are also very relevant (ie. False positives and false negatives). Rather than stating that their model "outperforms Megalodon at all coverages", it is relevant to know if it is either more accurate, more sensitive, or both.

- It is unclear how the datasets were divided to properly perform cross-validation. Which data was used for training, validation and testing is very important in any machine learning task, over the whole manuscript it is very difficult to follow how this splitting of the data was done. I advise the authors to perform chromosomal cross-validation, where all chromosomes but (e.g.) two are used for training, one for validation and one for testing. This approach is easy to implement, understand, and avoids any potential information leakage between datasets.

- In the Train models section it is mentioned: "The model parameters with the current best performance on the validation dataset are saved in every epoch". If the reported performance values are also based on this dataset there is an important leak of information in the cross-validation and the performance results could be overestimated.

Minor concerns:

- We highly recommend revising the document with the assistance of a native English speaker, the entire manuscript contains grammatical mistakes. Here are some examples of sentences that are grammatically wrong, very difficult to read or where it is unclear what is meant:

-- We develop a denoising process to train the tool to achieve high correlations with bisulfite sequencing for the detection of three contexts of 5mCs in plants.

-- Therefore, the detection of genome-wide CHG and CHH methylation is the same important as the detection of CpG methylation in plants.

-- The statistic method using the early version of Pacbio SMRT data to detect 5mCs exists low signal-to-noise ratio problem.

-- ..., which makes DNA degradation and amplification biases be avoided.

-- Furthermore, we find that the types of differentially methylated cytosines in repeat pairs show species-specific.

-- The differentially methylated repeat pairs in two replicates of *O. sativa* show great inconsistency, which implies that the differentially methylated repeat pairs are stable in spices.

-- Because fully methylated cytosines are much less than fully unmethylated cytosines, especially for CHH, it is difficult to collect positive training samples and results in an unbalanced training dataset.

- Along the manuscript, the following notation is used several times " $\sim 116\times$ coverage of reads". This is used to mention the amount of data used for training and validation in several places. However, it is unclear what the authors mean by this and it is difficult to really understand how much data is used in each case. Probably they indicate mean or median genome coverage, but this is not specified.

- A cytosine is either methylated, or unmethylated. There is no such thing as more or less methylated cytosines, instead there are cytosines that are more frequently methylated in an overall population, or less frequently methylated in a population. Aside from that, a region can be highly methylated, if multiple cytosines are methylated. However, a single cytosine can not be highly or lowly methylated. In several places, this is ambiguously phrased, for example:

"This may be due to the relatively less methylated sites and k-mers of CHH motif in *A. thaliana* (Supplementary Tables 3-4)."

"Megalodon tends to underpredict highly and intermediately methylated cytosines,"

- Some parameters of the model are poorly explained in the main text; for example:

"For each targeted 5mC site, DeepSignal-plant constructs four k-length features of the k-mer"

It is only apparent from figure 1 what those features are (signal, length, SD and base).

- The overview from figure 1 is not comprehensible without prior knowledge of deep-learning algorithms. Terms such as softmax, full connection or bidirectional long short-term memory layer are not commonly known by biologists. It may be appropriate to include 1-sentence introduction to these concepts.

- The authors state that they train 3 different models, one for each methylation context. It would be nice to see if the three models could be combined into 1 for several reasons:

- Effect on performance, it could be that the information from one context is relevant to predict methylation on another context. This could potentially improve performance.

- Using the models in a real-life application, one would require to detect methylation 3 times on one dataset, that is of course more computationally expensive than just running a single model.

- In Algorithm 2 the number of epochs (E) is mentioned as a parameter, but such parameter is not in the description of the algorithm itself.

- In Suppl.Fig.3 the diagram indicates that the samples are balanced and then the denoising procedure is done. Since the denoising procedure will change the label of some samples, the datasets might be slightly unbalanced again. Perhaps doing a second round of balance should be done after the denoising.

- A similar diagram for the training of Megalodon would make things clear in the comparison of the training of the two approaches.

- Hyperparameter optimization is a bit lacking. There is optimization on the length of the k-mers. The number of signals is not optimized, as a value is chosen based on a distribution. There is no mention on the optimization on the number of layers, learning rate, number of hidden dimensions, etc. The amount of hidden units in the layers is not mentioned at all. Reproducing the same model architecture is not possible without looking at the code.

- There is no mention regarding how fast can their models predict 5mC. This can be an important factor, especially since one has to basecall the data 3 times (one for each 5mC context).

- The authors move to the analysis of repeat pairs, but there is no explanation why repeat pairs are interesting or relevant to the field. Readers are left guessing why it is of importance to detect (differential) methylation in this context.

Regarding their github page:

- The code files in general lack comments

- The readme file looks very complete and descriptive, the authors have done a considerable effort here. Well done.

Reviewer #2 (Remarks to the Author):

In this manuscript, the authors developed deepsignal-plant, a deep learning tool for detection of DNA modification. There are many good tools for DNA methylation detection of CpG context from nanopore sequencing data, such as deepsignal developed by the authors previously and nanopolish. However, only a few tools which can detect CHG and CHH contexts are available. Their deepsignal-plant can detect methylation of those contexts. They performed Nanopore sequencing and bisulfite sequencing of *A. thaliana* and *O. sativa* and trained a bidirectional recurrent neural network (BRNN) with LSTM utilizing these data. Per site methylation rate predicted by deepsignal-plant showed higher pearson's

correlations with bisulfite sequencing than Megalodon. They also performed the cross-species evaluation of methylation calling using nanopore sequencing data and bisulfite sequencing data of *B. nigra*. Furthermore, they indicated that deepsignal-plant could cover the regions which bisulfite sequencing couldn't cover, and analyzed the differential methylation between repeat pair, which has same sequence but locate in different genomic coordinates.

Although there are some concerns as below, I assume that deepsignal plant is useful for plant research community.

Major

1. I believe the primary advantage of methylation calling from nanopore sequencing is that nanopore sequencing can determine the methylation patterns on long single DNA molecules. Therefore, it is important to evaluate the accuracy and the sensitivity at read level, as described in their previous paper (Ref. 1).

2. I could not understand the meaning of UpSet plots in Figures 4c, 4d and Supplementary Figure 21b. The detailed explanation about these figures should be added to Result section and Figure Legends.

3. I understand that genomic DNA of mammalian cells except for cells in early development and neural cells is rarely methylated in CHH and CHG contexts. However, some CHH and CHG sites are methylated. Although the deep learning-based tools might have a dependency on the datasets for its accurate detection, deepsignal-plant would be helpful for researchers of boarder research areas if it can be used also for mammalian dataset. Therefore, it is worth to evaluate whether deepsignal-plant can be used for mammalian dataset or not.

Minor

1. In this manuscript, there are many mistakes of tense and spelling (e.g. "sequence"->"sequenced" and "develop"->"developed" on page 2, "unmethylated 5mC" -> "unmethylated 5C" or "unmethylated C" on page 7). So, I strongly think that this manuscript should get English proofreading.

2. Many deep learning-based tools requires computational resources with high performance which it is not easy for many researchers to access. It is helpful to indicate the requirement of computational environment and the runtime for performing the deepsignal-plant.

3. In Method section of Bisulfite sequencing, the detailed information about the experiment for library preparation is not described. The authors should add the information about Bisulfite conversion kit and library preparation kit used for bisulfite sequencing.

4. It is shown that EM-seq can more uniformly cover the entire of genome, compared with bisulfite sequencing (ref. 2). It might be better to compare with EM-seq in the regard with covered sites, in addition to bisulfite sequencing.

5. In Figure 2, if training model by a combination of *A. thaliana* and *O.*, the author should state the fact more clearly. I think that the fact is important information to evaluate the performance of deepsignal-plant.

6. Though the definition of repeat pairs was described in Method section, it should be described also in Result section for readers.

7. In Figure 4ab, Supplementary Figure 21a and 22, they showed ratio of differentially methylated cytosines in repeat pairs, as label of x-axis. However, I assumed that x-axis means difference of methylation ratio between repeat pair. If wrong, they should explain the meaning in more detail in Result section and Legends.

The authors should add the explanation about dash lines (10%?) to Figure Legends.

Reference

1. Ni P, Huang N, Zhang Z, Wang DP, Liang F, Miao Y, Xiao CL, Luo F, Wang J. DeepSignal: detecting DNA methylation state from Nanopore sequencing reads using deep-learning. *Bioinformatics*. 2019 Nov 1;35(22):4586-4595.
2. Feng S, Zhong Z, Wang M, Jacobsen SE. Efficient and accurate determination of genome-wide DNA methylation patterns in *Arabidopsis thaliana* with enzymatic methyl sequencing. *Epigenetics Chromatin*. 2020 Oct 7;13(1):42.

Reviewer #3 (Remarks to the Author):

In this manuscript, the authors present a new tool, which they call "DeepSignal-plant", for the detection of cytosine DNA methylation in any context (i.e. CG, CHG and CHH, where H=A, T or C) using ONT (Nanopore) sequencing reads. So far, most cytosine methylation callers developed for Nanopore reads detect CG methylation only, the sole context of cytosine methylation in most mammalian cell types, including germ cells. However, plants methylate cytosines in all three contexts, which all contribute to the epigenetic control of transposable element (TE) and other repeat sequences and there many other groups of eukaryotes where non-CG methylation also plays important roles. Thus, being able to call reliably methylation at CHG and CHH in addition to CG sites using Nanopore sequencing reads is crucial for methylome studies in non-mammalian organisms.

The authors mainly focus on two plant species *A. thaliana* and *O. sativa* to carry out an in-depth evaluation of the performance of DeepSignal-plant. They first generate using the same DNA samples whole-genome bisulfite sequencing data, the gold standard for the detection of cytosine methylation in any context, and Nanopore sequencing data. Based on these two sources of data, they establish a carefully designed Nanopore data set for the training of a deep learning model, which they then evaluate using multiple comparisons with results obtained using BSseq.

Although this reviewer has no expertise in deep learning, the methodology presented as well as the numerous evaluation steps carried out appear properly justified and sound. In addition, the authors show that compared to Megalodon, recently developed by ONT and the only other existing tool that can detect methylated cytosines in all contexts, DeepSignal-plant is significantly better at calling CHH methylation and more consistent overall.

In sum, DeepSignal-plant fills an important gap in our ability to analyze cytosine methylation in all sequence contexts using Nanopore sequencing. For this reason, it should be of broad interest.

The manuscript in its present forms suffers from poor English, especially in the Introduction and need therefore careful language editing prior to being accepted for publication. Also, some of the methodology is difficult to follow for non-specialists. For instance, what does BRNN do; and what is the difference between signal features and inception blocks?

Other points:

In the introduction, there is some confusion in the way DNA methylation in *A. thaliana* is described: here it is the overall methylation level of CG, CHG and CHH sites that is reported, not the percentage of CGs, CHGs and CHHs that are methylated (at some level) across the genome. Lister et al, *Cell* 2008 should also be cited (in addition to ref 9). The sentences that follow on the different roles for CG, CHG and CHH methylation are also confusing and some of the statements are wrong.

We have pair sequenced...: replace with 'We have performed in parallel BSseq and Nanopore sequencing...'

In the last paragraph of the penultimate Results section, the sentence "Furthermore, the majority of those 5mCs are..." is unclear.

First sentence of the last Results section: replace "repeat pairs..." with "located within segmental

duplications...".

Summary

We appreciate the valuable comments and suggestions from the editor and reviewers. Based on the suggestions and comments from editor and reviewers, we revised our paper. We addressed those comments and suggestions carefully and included a point-by-point response below. We completely rewrote some paragraph and significant changes were highlighted by color.

Answers to Reviewer #1

Reviewer #1 (Remarks to the Author):

The authors describe a method to detect Cytosine methylation in the context of plants, where cytosine methylation occurs not only in CpG sites, but also in CHG and CHH trinucleotides. Existing (methylated) base callers have not been optimized for this context. They authors showcase the method in three plant models, and train and test their caller using bisulphite sequencing data.

They show that their method can produce data similar to bisulphite sequencing and detects methylation in sites that are not mappable using short read sequencing.

The detection of DNA modifications in a context unbiased manner is a very relevant research question. Research that helps improve models in this regard are very important and relevant for the Nanopore community.

Authors' Response. Thanks for the comments.

Major concerns:

- The authors show that the denoising method improves performance by improving the correct labelling of the samples. While this might be true, the approach itself looks like a self-fulfilling prophecy, as the same samples that are re-labelled are going to be used for training. A simulation experiment should be done to show that indeed the denoising method works as intended, this can be easily done by having a ground-truth dataset and purposely mislabelling several samples and evaluating whether the method would correct such mislabellings.

Authors' Response. Thanks for the above concern and suggestion! We perform the suggested simulation experiment using our *A. thaliana* sequencing data as follows:

(1) We first establish ground-truth datasets. Based on bisulfite sequencing, we select cytosines with methylation frequencies equal to 1 and 0. Then for each motif, we extracted corresponding true-positive and true-negative samples of the selected sites from Nanopore reads. We generate 9,388,125, 972,099, and 309,301 true-positive samples for CpG, CHG, and CHH, respectively. To establish a ground-truth dataset for each motif, we use the balancing method to generate balanced positive and negative training samples. (2) For each motif, we randomly change the labels of the certain number of negative samples from 0 (negative) to 1 (positive) in

the ground-truth dataset and remove the same number of true-positive samples with the mislabeled samples, to generate datasets with different mislabeled ratios (0%, 5%, 10%, 15%, and 20%). For example, in a dataset with a 10% mislabeled ratio, 10% of positive samples are mislabeled samples (*i.e.*, false-positive samples), while the total number of positive samples are still 9,388,125, 972,099, and 309,301 for CpG, CHG, and CHH, respectively. Then, we evaluate the denoising method using the datasets. We repeat 5 times the mislabeling-denoising experiment for each mislabel ratio.

The results show that, although a small portion of true-positive samples are removed, most of the mislabeled samples are removed by the denoising method. For example, in the datasets with a 10% mislabeled ratio, 15.3% (CG), 17.8% (CHG), 35.4% (CHH) true-positive samples are removed, while 96.9% (CG), 97.5% (CHG), 94.8% (CHH) mislabeled samples are removed.

We add a new paragraph in the revised manuscript and a section in Supplementary Note 1 to describe the simulation experiments.

- Nanopore sequencing combined with DeepSignal-plant detects many novel methylated sites in repeat regions. However there is no control to establish whether these calls are accurate or false positives. It could very well be possible that the kmers in repeat regions are significantly different from those in mappable regions, on which the algorithm was trained. The authors could show that there is significant overlap between the kmers in repeat regions, and the kmers used for training. If this overlap is not significant, this issue could for example be addressed by using synthetic controls, to validate that the model calls methylation correctly in previously unmappable regions.

Authors' Response. Thanks for the above concern and suggestion. As suggested, we analyze the k -mers ($k=13$) in the training dataset, and in the regions of *A. thaliana*, *O. sativa* (sample1 and sample2) which can only be covered by Nanopore sequencing for each motif. The result shows that 91.5%-97.0% CpG k -mers, 91.9%-94.8% CHG k -mers, and 80.3%-85.1% CHH k -mers in the previously unmappable regions are in the training dataset, which indicates that DeepSignal-plant could detect 5mC methylation accurately in those regions.

We have added this analysis in Supplementary Table 8 of the revised manuscript.

- The strategy employed to detect methylated cytosines uses a segmentation approach. This type of models was used in the past for basecalling, but nowadays state of the art basecallers do not require segmented data for prediction (eg. Chiron, Guppy, Bonito, Megalodon, ...). In this regard, it feels like this strategy is a step back regarding the technological advances of the past years.

-- Because of this, authors are bound to sample raw data datapoints for each base if there are more than their network input size allows.

-- This approach is also bound by the tool Tombo, future changes in the tool might affect performance of the models. This might also require re-training of the models for each Tombo version.

-- Finally, this also forces the need of having a reference genome available.

Authors' Response. We agree that DeepSignal-plant uses a segmentation approach for methylation calling, while basecalling tools (such as Chiron, Guppy, and Bonito) use Seq2Seq approaches to translate raw current signal sequences to nucleotide sequences. However, our results showed that the segmentation approach can achieve higher performance in 5mC methylation prediction, especially for the CHH (Fig. 2), which have a low methylation level in the genome and basecalling based methods could not optimize the methylation prediction for those sites.

If the number of raw signals is more than m ($m=16$ by default, which is larger than number of raw signals of 91.4% bases in our tests), DeepSignal-plant samples raw signals for each base. We have also performed hyperparameter tuning on m (Supplementary Fig. 30).

We agree that once the algorithm (dynamic time warping) for re-squiggle in tombo changes significantly, we need to re-train the models of DeepSignal-plant. In this study, we use the newest version of tombo (v1.5.1) for re-squiggle. Using $20\times$ (mean genome coverage) *A. thaliana* Nanopore data, we assess the re-squiggle results of two early versions of tombo (v1.0, v1.2). We find that compared to tombo v1.5.1, tombo v1.0 and v1.2 show a larger percentage of failed reads in the re-squiggle process. We also use DeepSignal-plant to call methylation from the re-squiggle results of these three tombo versions. As shown in the following figure, the effect of the re-squiggle tool on the performance of DeepSignal-plant is limited. We will regularly maintain DeepSignal-plant, and update the pre-trained models once we need to.

Fig. R1 Comparison of different versions of tombo. **a:** Percentage of failed reads in re-squiggle process. **b:** Performance of DeepSignal-plant on the results of different tombo versions. We compare cytosines with at least $5\times$ coverage reads after the resquiggle process of all tombo versions.

DeepSignal-plant needs a publicly available reference genome, or an assembly genome of the corresponding species/sample for calling 5mC methylation status, since the reference genome allows the better alignment of signals to the base, then results in high accuracy prediction.

- The authors compare their model with Megalodon. While it is true that megalodon can be trained to detect base modifications, Megalodon's task is also to do regular basecalling. For this reason, Megalodon's task is far more complex than DeepSignal-plant, as the latest only has to work on methylation detection.

-- It is mentioned that two Megalodon configurations are trained (methods section), but in the end only 3 models are trained. It is unclear which of the two configurations was used in the end.

-- It is unclear if the authors also employed their denoising approach to improve the dataset during Megalodon training.

-- It is unclear if the authors also employed their *k*-mer balancing approach for Megalodon training.

-- For these reasons, I believe the comparison to be a bit unfair towards Megalodon.

Authors' Response. We agree that Megalodon's task is more complex than DeepSignal-plant. Besides modified base calling, Megalodon is also capable of sequence variant calling, whereas DeepSignal-plant is only designed for methylation calling.

The two configuration files *res_dna_r941_prom_modbases_5mC_CpG_v001.cfg* and *res_dna_r941_min_modbases-all-context_v001.cfg* are two pre-trained model files which are got from Megalodon. In the revised manuscript, we have trained a single model for 5mC detection for Megalodon, rather than training 3 models (one for each motif). To train the new 5mC model, we used *res_dna_r941_min_modbases-all-context_v001.cfg* as the initial model. We have re-written the *Methods-Retrain Megalodon* section.

Megalodon employs a Seq2Seq model in Guppy for methylation status prediction (Supplementary Note 5 and Supplementary Fig. 4b). First, to train the Seq2Seq model, mapping between consecutive raw signal sequences and nucleotide sequences, rather than discrete sites with corresponding features and labels, are needed. Second, the training process of Megalodon allows to leverage cytosines of all methylation status for annotation, which gives better performance than just selecting unmethylated and fully methylated cytosines for annotation [1]. In summary, training the Seq2Seq model of Megalodon/Guppy does not need discrete samples that are labeled as positive (1) or negative (0). Therefore, we could not employ the denoising and *k*-mer balancing approaches for Megalodon training.

References

[1] Oxford Nanopore Technologies. *Megalodon*. https://nanoporetech.github.io/megalodon/modbase_training.html. Accessed 25 June 2021.

- The authors compare the output of their model to bisulphite sequencing and measure its performance using Pearson correlation. However, sensitivity and accuracy are

also very relevant (ie. False positives and false negatives). Rather than stating that their model “outperforms Megalodon at all coverages”, it is relevant to know if it is either more accurate, more sensitive, or both.

Authors' Response. Thanks for your suggestion. We evaluate DeepSignal-plant and Megalodon at read level using *A. thaliana*, *O. sativa* (sample1 and sample2) and *B. nigra* data. For each species, we selected cytosines with 1 and 0 methylation frequency based on bisulfite sequencing. Then we extract corresponding positive and negative samples of the selected sites from Nanopore reads for evaluation. The results show that DeepSignal-plant gets higher sensitivities than Megalodon for all motifs of all species. DeepSignal-plant also gets higher accuracies than Megalodon, except for the CpG motif of *B. nigra* (0.9257 vs 0.9394).

In the revised manuscript, we have added and discussed the results in Supplementary Table 7 and the *Results* section.

- *It is unclear how the datasets were divided to properly perform cross-validation. Which data was used for training, validation and testing is very important in any machine learning task, over the whole manuscript it is very difficult to follow how this splitting of the data was done. I advise the authors to perform chromosomal cross-validation, where all chromosomes but (e.g.) two are used for training, one for validation and one for testing. This approach is easy to implement, understand, and avoids any potential information leakage between datasets.*

Authors' Response. Thanks for the above concern and suggestion. In our original analysis, we use read-based-independent validation, in which $\sim 500\times$ (mean genome coverage) Nanopore reads of *A. thaliana* and $\sim 115\times$ reads of *O. sativa* (sample1) are used for training, while $\sim 100\times$ reads of *A. thaliana*, $\sim 100\times$ reads of *O. sativa* (sample1), $\sim 100\times$ reads of *O. sativa* (sample2), and $\sim 78\times$ reads of *B. nigra* are used for testing. To train a model of DeepSignal-plant, we extract samples from the reads used for training, of which 99% samples were used for model training, 1% samples were used for model validation. In the revised manuscript, we made this clearer in *Evaluation of DeepSignal-plant using Nanopore data of A. thaliana and O. sativa* of the *Results* section, *Methods* section, and Supplementary Table 10.

In the revised manuscript, using *A. thaliana* data ($\sim 500\times$ training reads and $\sim 20\times$ randomly selected testing reads), we perform a cross-chromosomal validation as suggested. We divide reads based on the chromosomes that they are mapped to. Then we extract samples from the reads aligned to chr1-3 for model training, and extract samples from the reads aligned to chr4 for model validation. The reads aligned to chr5 are used for testing. In this chromosomal cross-validation, DeepSignal-plant got high correlations with bisulfite sequencing. We have added this analysis in the *Evaluation of DeepSignal-plant using Nanopore data of A. thaliana and O. sativa* of *Results* section and Supplementary Fig. 7 of the revised manuscript.

- *In the Train models section it is mentioned: "The model parameters with the current best performance on the validation dataset are saved in every epoch". If the reported performance values are also based on this dataset there is an important leak of*

information in the cross-validation and the performance results could be overestimated.

Authors' Response. As mentioned above, for the Nanopore data of *A. thaliana* and *O. sativa* (sample1), we first divide the reads into two groups: reads for training, and reads for testing, respectively. To train a model of DeepSignal-plant, we extract samples from the reads used for training. The extracted samples are processed by the balancing and denoising method and then divided into training dataset and validation dataset at a ratio of 99:1. After training, we evaluate the model using the reads for testing. All Nanopore reads of *O. sativa* (sample2) and *B. nigra* were used only for testing.

We have made this clear in the *Methods* section of the revised manuscript.

Minor concerns:

- We highly recommend revising the document with the assistance of a native English speaker; the entire manuscript contains grammatical mistakes. Here are some examples of sentences that are grammatically wrong, very difficult to read or where it is unclear what is meant:

-- We develop a denoising process to train the tool to achieve high correlations with bisulfite sequencing for the detection of three contexts of 5mCs in plants.

-- Therefore, the detection of genome-wide CHG and CHH methylation is the same important as the detection of CpG methylation in plants.

-- The statistic method using the early version of Pacbio SMRT data to detect 5mCs exists low signal-to-noise ratio problem.

-- ..., which makes DNA degradation and amplification biases be avoided.

-- Furthermore, we find that the types of differentially methylated cytosines in repeat pairs show species-specific.

*-- The differentially methylated repeat pairs in two replicates of *O. sativa* show great inconsistency, which implies that the differentially methylated repeat pairs are stable in species.*

-- Because fully methylated cytosines are much less than fully unmethylated cytosines, especially for CHH, it is difficult to collect positive training samples and results in an unbalanced training dataset.

Authors' Response. We have read the paper carefully and corrected all orthographic and grammatical mistakes accordingly.

- Along the manuscript, the following notation is used several times "~116× coverage of reads". This is used to mention the amount of data used for training and validation in several places. However, it is unclear what the authors mean by this and it is difficult to really understand how much data is used in each case. Probably they indicate mean or median genome coverage, but this is not specified.

Authors' Response. We apologize for the confusion. By using "×" we mean "mean genome coverage". We have emphasized this in the revised manuscript.

- A cytosine is either methylated, or unmethylated. There is no such thing as more or less methylated cytosines, instead there are cytosines that are more frequently methylated in an overall population, or less frequently methylated in a population. Aside from that, a region can be highly methylated, if multiple cytosines are methylated. However, a single cytosine can not be highly or lowly methylated. In several places, this is ambiguously phrased, for example:

*“This may be due to the relatively less methylated sites and k-mers of CHH motif in *A. thaliana* (Supplementary Tables 3-4).”*

“Megalodon tends to underpredict highly and intermediately methylated cytosines,”

Authors' Response. Thanks for pointing this out. We have read the manuscript carefully and corrected all relative sentences.

- Some parameters of the model are poorly explained in the main text; for example:

“For each targeted 5mC site, DeepSignal-plant constructs four k-length features of the k-mer”

It is only apparent from figure 1 what those features are (signal, length, SD and base).

Authors' Response. We have re-written the description of the model to make it clearer in *The DeepSignal-plant algorithm and training process* of Results section of the revised manuscript.

- The overview from figure 1 is not comprehensible without prior knowledge of deep-learning algorithms. Terms such as softmax, full connection or bidirectional long short-term memory layer are not commonly known by biologists. It may be appropriate to include 1-sentence introduction to these concepts.

Authors' Response. Thanks for the suggestion. We have added more descriptions of the neural network units (softmax, full connection and bidirectional long short-term memory layer) in the caption of Fig. 1.

- The authors state that they train 3 different models, one for each methylation context. It would be nice to see if the three models could be combined into 1 for several reasons:

-- Effect on performance, it could be that the information from one context is relevant to predict methylation on another context. This could potentially improve performance.

-- Using the models in a real-life application, one would require to detect methylation 3 times on one dataset, that is of course more computationally expensive than just running a single model.

Authors' Response. Thanks for the suggestion! We combine the training datasets of 3 motifs and train only one 5mC model of DeepSignal-plant as suggested. The 5mC model outperformed the original three models for CpG/CHH/CHG detection. We also train a 5mC model of Megalodon, which also got higher performances for CHG and CHH:

Fig. R2 Comparison of motif-specific models and motif-combined models of DeepSignal-plant and Megalodon on 20× *A. thaliana* reads. Models of DeepSignal-plant and Megalodon were trained using combined reads of *A. thaliana* and *O. sativa*.

We have then used the one 5mC model for downstream analysis. In the revised manuscript, we add Supplementary Fig. 6, update Fig. 2, Fig. 3a-b, Fig. 4, Supplementary Fig. 8-15, Supplementary Fig. 17, Supplementary Fig. 21, Supplementary Fig. 24-28, Supplementary Table 5, 6 and 9, and the relevant discussion.

- In Algorithm 2 the number of epochs (E) is mentioned as a parameter, but such parameter is not in the description of the algorithm itself.

Authors' Response. We have added E in line 6 and line 8 of Algorithm 2.

- In Suppl.Fig.3 the diagram indicates that the samples are balanced and then the denoising procedure is done. Since the denoising procedure will change the label of some samples, the datasets might be slightly unbalanced again. Perhaps doing a second round of balance should be done after the denoising.

-- A similar diagram for the training of Megalodon would make things clear in the comparison of the training of the two approaches.

Authors' Response. Thanks for your suggestion. In the denoising method, we balance the samples after each iteration of the denoising procedure (line 15, Algorithm 2). We made this clearer in *Denoise training samples* of *Methods* section.

We have added a diagram for the training of Megalodon in Supplementary Fig. 4b. A detailed description of training and methylation calling using Megalodon is in Supplementary Note 5.

- Hyperparameter optimization is a bit lacking. There is optimization on the length of the k -mers. The number of signals is not optimized, as a value is chosen based on a distribution. There is no mention on the optimization on the number of layers,

learning rate, number of hidden dimensions, etc. The amount of hidden units in the layers is not mentioned at all. Reproducing the same model architecture is not possible without looking at the code.

Authors' Response. Using *A. thaliana* data, we perform hyperparameter tuning on the number of signals, the number of layers, learning rate, the number of hidden units in BiLSTM as suggested. The analysis is added in Supplementary Fig. 30-31, Supplementary Table 11, and Supplementary Note 3.

- There is no mention regarding how fast can their models predict 5mC. This can be an important factor, especially since one has to basecall the data 3 times (one for each 5mC context).

Authors' Response. We evaluated the running time and peak memory of three main steps in the pipeline of DeepSignal-plant: (1) Basecall using Guppy; (2) Re-squiggle using Tombo; (3) Call methylation using DeepSignal-plant. The data used for evaluation include 100× (mean genome coverage) *A. thaliana* Nanopore reads, 100× *O. sativa* (sample1) Nanopore reads, 100× *O. sativa* (sample2) Nanopore reads, and 78× *B. nigra* Nanopore reads. We processed all data at a server with 40 CPU processors (Intel(R) Xeon(R) CPU E5-2676 v3 @ 2.40GHz), 256 GB RAM, and a 12GB TITAN X (Pascal) GPU.

We add this analysis in Supplementary Note 4 and Supplementary Table 12 of the revised manuscript.

- The authors move to the analysis of repeat pairs, but there is no explanation why repeat pairs are interesting or relevant to the field. Readers are left guessing why it is of importance to detect (differential) methylation in this context.

Authors' Response. Thanks for your concern. By analyzing differential methylation in repeats, we hope that novel insights could be provided to the correlation between duplicate gene transcription and methylation signatures [1]. Furthermore, as paralogous sequence variants (PSVs) have been proved to be capable of resolving segmental duplications [2], we hope that the differential methylation between repeat pairs could also be helpful to resolve collapsed regions of segmental duplications in de novo assemblies of plants.

Reference:

[1] Vollger MR, Guitart X, Dishuck PC, Mercuri L, Harvey WT, Gershman A, Diekhans M, Sulovari A, Munson KM, Lewis AM, Hoekzema K. Segmental duplications and their variation in a complete human genome. bioRxiv. 2021 Jan 1.

[2] Vollger MR, Dishuck PC, Sorensen M, Welch AE, Dang V, Dougherty ML, Graves-Lindsay TA, Wilson RK, Chaisson MJ, Eichler EE. Long-read sequence and assembly of segmental duplications. Nature methods. 2019 Jan;16(1):88-94.

Regarding their github page:

- The code files in general lack comments

- The readme file looks very complete and descriptive, the authors have done a considerable effort here. Well done.

Authors' Response. Thanks for the comments. We have revised the code files and added more comments.

Answers to Reviewer #2

Reviewer #2 (Remarks to the Author):

*In this manuscript, the authors developed deepsignal-plant, a deep learning tool for detection of DNA modification. There are many good tools for DNA methylation detection of CpG context from nanopore sequencing data, such as deepsignal developed by the authors previously and nanopolish. However, only a few tools which can detect CHG and CHH contexts are available. Their deepsignal-plant can detect methylation of those contexts. They performed Nanopore sequencing and bisulfite sequencing of *A. thaliana* and *O. sativa* and trained a bidirectional recurrent neural network (BRNN) with LSTM utilizing these data. Per site methylation rate predicted by deepsignal-plant showed higher pearson's correlations with bisulfite sequencing than Megalodon. They also performed the cross-species evaluation of methylation calling using nanopore sequencing data and bisulfite sequencing data of *B. nigra*. Furthermore, they indicated that deepsignal-plant could cover the regions which bisulfite*

sequencing couldn't cover, and analyzed the differential methylation between repeat pair, which has same sequence but locate in different genomic coordinates.

Although there are some concerns as below, I assume that deepsignal plant is useful for plant research community.

Authors' Response. Thanks for the positive comments.

Major

1. I believe the primary advantage of methylation calling from nanopore sequencing is that nanopore sequencing can determine the methylation patterns on long single DNA molecules. Therefore, it is important to evaluate the accuracy and the sensitivity at read level, as described in their previous paper (Ref. 1).

Authors' Response. Thanks for your suggestion. As suggested, we evaluate DeepSignal-plant and Megalodon at read level using *A. thaliana*, *O. sativa* (sample1 and sample2), and *B. nigra* data. For each species, we select cytosines with 1 and 0 methylation frequency based on bisulfite sequencing. Then we extract corresponding positive and negative samples of the selected sites from Nanopore reads for evaluation. The results show that DeepSignal-plant gets higher accuracies than Megalodon, except for the CpG motif of *B. nigra* (0.9257 vs 0.9394). DeepSignal-plant also gets higher sensitivities than Megalodon for all motifs of all species, while Megalodon got higher specificities.

In the revised manuscript, we have added and discussed the results in Supplementary Table 7 and the *Results* section.

2. *I could not understand the meaning of UpSet plots in Figures 4c, 4d and Supplementary Figure 21b. The detailed explanation about these figures should be added to Result section and Figure Legends.*

Authors' Response. In this study, suppose the methylation frequencies of a cytosine in the same relative position of the repeat pair are $rmet_1$ and $rmet_2$, the cytosine is said to be differentially methylated if $|rmet_1 - rmet_2| \geq 0.5$. A repeat pair is said to be differentially methylated if there are at least 10% cytosines (or CpG sites, CHG sites, CHH sites independently) that are differentially methylated in the repeat pair. For each species/sample, we generate four sets of differentially methylated repeat pairs based on cytosines, CpGs, CHGs and CHHs, respectively. Then, we compare the intersection of the differentially methylated repeat pairs among the four sets using UpSet plot.

In each UpSet plot, circles below in each column indicate sets that are part of the intersection (*i.e.*, corresponding one segment in a Venn diagram). The up bars indicate the size of each intersection. The left bars indicate the total size of each set.

We have added more explanation in the *Results* section and caption of the figures (Fig. 4c-d and Supplementary Fig. 24b) in the revised manuscript.

3. *I understand that genomic DNA of mammalian cells except for cells in early development and neural cells is rarely methylated in CHH and CHG contexts. However, some CHH and CHG sites are methylated. Although the deep learning-based tools might have a dependency on the datasets for its accurate detection, deepsignal-plant would be helpful for researchers of boarder research areas if it can be used also for mammalian dataset. Therefore, it is worth to evaluate whether deepsignal-plant can be used for mammalian dataset or not.*

Authors' Response. Thanks for the suggestion. We will find biologist collaborators who can provide mammalian samples for us to sequene. We will add the results to our GitHub site in the future.

Minor

1. *In this manuscript, there are many mistakes of tense and spelling (e.g. "sequence"->"sequenced" and "develop"->"developed" on page 2, "unmethylated 5mC" -> "unmethylated 5C" or "unmethylated C" on page 7). So, I strongly think that this manuscript should get English proofreading.*

Authors' Response. Thanks for pointing out the mistakes! We have used the "present tense" in the manuscript and make it consistent. We have corrected the mistakes accordingly and revised the whole manuscript.

2. *Many deep learning-based tools requires computational resources with high performance which it is not easy for many researchers to access. It is helpful to indicate the requirement of computational environment and the runtime for*

performing the deepsignal-plant.

Authors' Response. Thanks for your suggestion. We evaluate the running time and peak memory of three main steps in the pipeline of DeepSignal-plant: (1) Basecall using Guppy; (2) Re-squiggle using Tombo; (3) Call methylation using DeepSignal-plant. The data used for evaluation include 100× (mean genome coverage) *A. thaliana* Nanopore reads, 100× *O. sativa* (sample1) Nanopore reads, 100× *O. sativa* (sample2) Nanopore reads, and 78× *B. nigra* Nanopore reads. We process all data at a server with 40 CPU processors (Intel(R) Xeon(R) CPU E5-2676 v3 @ 2.40GHz), 256 GB RAM, and a 12GB TITAN X (Pascal) GPU.

We add this analysis in Supplementary Note 4 and Supplementary Table 12 of the revised manuscript.

3. In Method section of Bisulfite sequencing, the detailed information about the experiment for library preparation is not described. The authors should add the information about Bisulfite conversion kit and library preparation kit used for bisulfite sequencing.

Authors' Response. We use TIANGEN DNA Bisulfite Conversion Kit (cat #: DP215, TIANGEN BIOTECH) and TruSeq DNA Methylation Kit (cat #: EGMK91324, Illumina) as bisulfite conversion kit and library preparation kit, respectively, for sequencing three technical replicates of *A. thaliana* and *O. sativa* (sample2). For *O. sativa* (sample1), the bisulfite conversion kit and library preparation kit are EZ DNA Methylation-Gold Kit (Zymo Research) and MGIEasy Whole Genome Bisulfite Sequencing Library Prep Kit (16 RXN) (BGI), respectively. We have added the information in the *Method section of Bisulfite sequencing*.

4. It is shown that EM-seq can more uniformly cover the entire of genome, compared with bisulfite sequencing (ref. 2). It might be better to compare with EM-seq in the regard with covered sites, in addition to bisulfite sequencing.

Authors' Response. Thanks for your suggestion. As suggested, we download 8 EM-seq replicates of *A. thaliana* from ref. 2, which are about 370× (mean genome coverage) reads in total (Table R1). We also download the corresponding BS-seq data of the 8 replicates from ref. 2. We process the EM-seq and BS-seq data with Bismark (v0.20.0). For comparison, we counted all cytosines which are covered with at least 5 reads in at least 1 replicate.

We first compare the covered sites between the 8 EM-seq replicates and the corresponding BS-seq replicates from ref. 2 (Fig. R3a). The results show that the EM-seq replicates covered more cytosines than the BS-seq replicates. We then compare the cytosines covered by the EM-seq replicates, the 3 BS-seq replicates (~116×, ~131×, and ~116×, respectively) and the Nanopore data (Fig. R3b). From the results, we observe that Nanopore sequencing covered more cytosines than both BS-seq and EM-seq.

Table R1. EM-seq and BS-seq replicates downloaded from ref. 2.

ID	replicate	mean genome coverage	
		EM-seq	BS-seq

SRR11906626	Flower-4-50ng-18PCR	58.9	22.8
SRR11906602	Flower-3-50ng-18PCR	51.0	27.3
SRR11906614	Flower-4-25ng-18PCR	49.1	23.4
SRR11906590	Flower-3-25ng-18PCR	47.2	27.9
SRR11906608	Flower-4-150ng-18PCR	47.0	28.9
SRR11906584	Flower-3-150ng-18PCR	44.5	26.4
SRR11906596	Flower-3-400ng-18PCR	39.4	34.9
SRR11906620	Flower-4-400ng-18PCR	37.6	37.1

Fig. R3 Comparison of cytosines covered by EM-seq, BS-seq and Nanopore sequencing. **a:** Comparison of cytosines covered by 8 EM-seq replicates and 8 BS-seq replicates from ref. 2. **b:** Comparison of cytosines covered by 8 EM-seq replicates from ref. 2, 3 BS-seq replicates and Nanopore sequencing data used in our work.

5. In Figure 2, if training model by a combination of *A. thaliana* and *O.*, the author should state the fact more clearly. I think that the fact is important information to evaluate the performance of deepsignal-plant.

Authors' Response. Thanks for the suggestion. In the revised manuscript, we have stated the fact more clearly in *Comparison of DeepSignal-plant to other tools for 5mC detection* of the Results section, and the caption of Fig. 2.

6. Though the definition of repeat pairs was described in Method section, it should be described also in the Result section for readers.

Authors' Response. We have added the definition of repeat pairs in *Differentially methylated cytosines in repeat pairs* of Result section of the revised manuscript.

7. In Figure 4ab, Supplementary Figure 21a and 22, they showed ratio of differentially methylated cytosines in repeat pairs, as label of x-axis. However, I

assumed that x-axis means difference of methylation ratio between repeat pair. If wrong, they should explain the meaning in more detail in Result section and Legends.

The authors should add the explanation about dash lines (10%?) to Figure Legends.

Authors' Response. The label of x-axis “Ratio of differentially methylated cytosines” means the ratio of differentially methylated cytosines to total cytosines in one repeat pair. In the revised manuscript, we have added more description in *Differentially methylated cytosines in repeat pairs* of Results section and the legends of figures (Fig. 4, Supplementary Fig. 24 and 25).

We add dash lines in $x=10\%$, as we treat repeat pairs in which there are at least 10% cytosines that are differentially methylated as differentially methylated repeat pairs. We have added an explanation of the dash lines in the legends of figures (Fig. 4, Supplementary Fig. 24 and 25).

Reference

1. Ni P, Huang N, Zhang Z, Wang DP, Liang F, Miao Y, Xiao CL, Luo F, Wang J. DeepSignal: detecting DNA methylation state from Nanopore sequencing reads using deep-learning. *Bioinformatics*. 2019 Nov 1;35(22):4586-4595.
2. Feng S, Zhong Z, Wang M, Jacobsen SE. Efficient and accurate determination of genome-wide DNA methylation patterns in *Arabidopsis thaliana* with enzymatic methyl sequencing. *Epigenetics Chromatin*. 2020 Oct 7;13(1):42.

Answers to Reviewer #3

Reviewer #3 (Remarks to the Author):

In this manuscript, the authors present a new tool, which they call "DeepSignal-plant", for the detection of cytosine DNA methylation in any context (i.e. CG, CHG and CHH, where H=A, T or C) using ONT (Nanopore) sequencing reads. So far, most cytosine methylation callers developed for Nanopore reads detect CG methylation only, the sole context of cytosine methylation in most mammalian cell types, including germ cells. However, plants methylate cytosines in all three contexts, which all contribute to the epigenetic control of transposable element (TE) and other repeat sequences and there many other groups of eukaryotes where non-CG methylation also plays important roles. Thus, being able to call reliably methylation at CHG and CHH in addition to CG sites using Nanopore sequencing reads is crucial for methylome studies in non-mammalian organisms.

*The authors mainly focus on two plant species *A. thaliana* and *O. sativa* to carry out an in-depth evaluation of the performance of DeepSignal-plant. They first generate using the same DNA samples whole-genome bisulfite sequencing data, the gold standard for the detection of cytosine methylation in any context, and Nanopore sequencing data. Based on these two sources of data, they establish a carefully*

designed Nanopore data set for the training of a deep learning model, which they then evaluate using multiple comparisons with results obtained using BSseq.

Although this reviewer has no expertise in deep learning, the methodology presented as well as the numerous evaluation steps carried out appear properly justified and sound. In addition, the authors show that compared to Megalodon, recently developed by ONT and the only other existing tool that can detect methylated cytosines in all contexts, DeepSignal-plant is significantly better at calling CHH methylation and more consistent overall.

In sum, DeepSignal-plant fills an important gap in our ability to analyze cytosine methylation in all sequence contexts using Nanopore sequencing. For this reason, it should be of broad interest.

Authors' Response. Thanks for the positive comments.

The manuscript in its present forms suffers from poor English, especially in the Introduction and need therefore careful language editing prior to being accepted for publication. Also, some of the methodology is difficult to follow for non-specialists. For instance, what does BRNN do; and what is the difference between signal features and inception blocks?

Authors' Response. Thanks for the suggestion. We have read the paper carefully and corrected all orthographic and grammatical mistakes accordingly. We made the methodology clearer in *The DeepSignal-plant algorithm and training process* of Results section, *The framework of DeepSignal-plant* of Methods section and Supplementary Note 2.

A BRNN is a neural network model for sequential data. Each BRNN includes a forward RNN and a backward RNN to catch both forward and backward context. A RNN scans the sequence of data and encodes the sequential information into a latent representation. In DeepSignal-plant, BRNN is used to process sequence features and signal features of a cytosine. We add more description in the *Model architecture* of Method section. More detail about BRNN model is in Supplemental Note 2.

Inception block is a neural network architecture which are composed of convolutional neural networks. Signal features of a cytosine are a $k \times m$ matrix ($k=13$, $m=16$) which contains signal values extracted from the corresponding Nanopore read of the cytosine. In DeepSignal [1], inception blocks are used to process signal features. In DeepSignal-plant, BRNN is used to process both signal features and sequence features, which reduce the model size to one-eighth of previous DeepSignal. In the revised manuscript, we have rewrote the sentence to make it clear as “By using BRNN to process both signal features and sequence features, the size of the DeepSignal-plant model is only one-eighth of the size of DeepSignal (Supplementary Table 2).”

We also added more descriptions of the neural network units (softmax, full connection and BiLSTM) in the caption of Fig. 1.

References

[1] Ni P, Huang N, Zhang Z, Wang DP, Liang F, Miao Y, Xiao CL, Luo F, Wang J.

DeepSignal: detecting DNA methylation state from Nanopore sequencing reads using deep-learning. *Bioinformatics*. 2019 Nov 1;35(22):4586-4595.

Other points:

*In the introduction, there is some confusion in the way DNA methylation in *A. thaliana* is described: here it is the overall methylation level of CG, CHG and CHH sites that is reported, not the percentage of CGs, CHGs and CHHs that are methylated (at some level) across the genome. Lister et al, Cell 2008 should also be cited (in addition to ref 9). The sentences that follow on the different roles for CG, CHG and CHH methylation are also confusing and some of the statements are wrong.*

Authors' Response. We apologize for the confusion. By using “overall methylation level”, we mean the percentage of cytosines that are methylated at read level, which is the number of methylated cytosines divided by the number of total cytosines in whole sequenced reads [1][2]. We have corrected the improper expression to “there are 24% CpG, 6.7% CHG and 1.7% CHH methylated at read level in *Arabidopsis thaliana*, while *Beta vulgaris* has 92.5% CpG, 81.2% CHG, and 18.8% CHH being methylated at read level”. We cited “Lister et al, Cell 2008” as suggested.

We have re-written the sentences about the different roles of CG, CHG, and CHH methylation.

References

[1] Cokus SJ, Feng S, Zhang X, Chen Z, Merriman B, Haudenschild CD, Pradhan S, Nelson SF, Pellegrini M, Jacobsen SE. Shotgun bisulphite sequencing of the *Arabidopsis* genome reveals DNA methylation patterning. *Nature*. 2008 Mar;452(7184):215-9.

[2] Niederhuth CE, Bewick AJ, Ji L, Alabady MS, Do Kim K, Li Q, Rohr NA, Rambani A, Burke JM, Udall JA, Egesi C. Widespread natural variation of DNA methylation within angiosperms. *Genome biology*. 2016 Dec;17(1):1-9.

We have pair sequenced...: replace with 'We have performed in parallel BSseq and Nanopore sequencing...

In the last paragraph of the penultimate Results section, the sentence "Furthermore, the majority of those 5mCs are..." is unclear.

First sentence of the last Results section: replace "repeat pairs..." with "located within segmental duplications..."

Authors' Response. Thanks for pointing out the mistakes. We have corrected the mistakes accordingly.

REVIEWERS' COMMENTS

Reviewer #1 (Remarks to the Author):

The authors have done a tremendous job answering the questions. I have no further comments.

Reviewer #2 (Remarks to the Author):

The authors have addressed most of my comments. The manuscript improved well.

Minor point:

I feel that the manuscript might need to be checked by a native English speaker, as there are still problems with the English language. For examples, I found some misspelling as follows: "sanple"->"sample" in Supplementary table 12, and "tRAN"-> "tRNA" in the Discussion section p.15 l.4.

Reviewer #3 (Remarks to the Author):

The authors have addressed satisfactorily all of the points raised by me and the other reviewers. The revised version manuscript is much improved compared to the original submission.

Summary

We appreciate the valuable comments and suggestions from the editor and reviewers. Based on the suggestions and comments from the editor and reviewers, we revised our paper. We addressed those comments and suggestions carefully.

Answers to Reviewer #1

Reviewer #1 (Remarks to the Author):

The authors have done a tremendous job answering the questions. I have no further comments.

Authors' Response. We appreciate the reviewer for the positive and supportive comment.

Answers to Reviewer #2

Reviewer #2 (Remarks to the Author):

The authors have addressed most of my comments. The manuscript improved well.

Authors' Response. We appreciate the reviewer for the positive and supportive comment.

Minior point:

I feel that the manuscript might need to be checked by a native English speaker, as there are still problems with the English language. For examples, I found some misspelling as follows: "sanple"->"sample" in Supplementary table 12, and "tRAN"-> "tRNA" in the Discussion section p.15 l.4.

Authors' Response. Thanks for the suggestion! We have corrected the typos accordingly and read the paper carefully to correct all orthographic and grammatical mistakes.

Answers to Reviewer #3

Reviewer #3 (Remarks to the Author):

The authors have addressed satisfactorily all of the points raised by me and the other reviewers. The revised version manuscript is much improved compared to the original submission.

Authors' Response. We appreciate the reviewer for the positive and supportive comment.